# Order Matters: Unveiling the Hidden Impact of Macro Placement Sequences via Proxy-Guided LLM Evolution

**Shibing Mo** [1 2]  **Jing Liu** [* 1 2]  **Jianchu Xu** [1]  **Ruilin Wu** [2]

## Abstract

Macro placement is a fundamental step in modern chip physical design, playing a crucial role in determining the solution quality of high-dimensional combinatorial optimization problems. Despite recent advancements in machine learning for spatial coordinate determination, the temporal dimension of placement sequencing remains largely governed by static heuristics. In this work, we demonstrate that the placement sequence is not merely a preprocessing step but a decisive factor in optimization, where suboptimal early decisions trigger irreversible domino effects that constrain the solution space. To harness this unexplored dimension, we propose **OrderPlace**, a proxy-guided LLM evolution framework for automatically discovering macro placement order strategies. Instead of relying on manually crafted heuristics such as area- or connectivity-based ordering, OrderPlace explores a broader space of code-level policies, ranging from static scoring metrics to dynamic physics-inspired mechanisms. To mitigate the prohibitive cost of evaluating sequences, we introduce a lightweight proxy evaluation mechanism that efficiently filters candidates using a deterministic greedy probe. Experimental results on the standard ISPD 2005 benchmarks demonstrate that OrderPlace discovers novel ordering strategies. Compared with WireMask-EA and the state-of-the-art method EGPlace, OrderPlace reduces wirelength by 34.04% and 14.08%, respectively.

* indicates the corresponding author. [1]the School of Artifcial Intelligence, Xidian University [2]the Guangzhou Institute of Technology, Xidian University. Correspondence to: Shibing Mo <msb@stu.xidian.edu.cn>, Jing Liu <neouma@mail.xidian.edu.cn>.

*Proceedings of the 43$^{rd}$ International Conference on Machine Learning*, Seoul, South Korea. PMLR 306, 2026. Copyright 2026 by the author(s).

## 1. Introduction

Macro placement stands as a cornerstone in modern chip design, fundamentally dictating the quality, performance, and manufacturability of the final design (MacMillen et al., 2000; Kim & Markov, 2012). Formally, this is a large-scale combinatorial optimization problem that requires determining the physical positions of macros within a fixed chip canvas without any overlap (Wang et al., 2009). The task is characterized by high-dimensional constraints and extreme sensitivity to boundary conditions. Despite recent advances in machine learning that have accelerated this process (Mirhoseini et al., 2021; Lin et al., 2019), macro placement remains a formidable challenge and an underexplored frontier in electronic design automation due to the immense design space (Majumdar et al., 2024).

Recent machine learning methods have provided new perspectives for tackling this challenge, evolving from constructive approaches (Shi et al., 2023; Deng et al., 2025) to reinforcement learning (RL) agents (Lai et al., 2022; Geng et al., 2024) and transformer-based architectures (Lai et al., 2023). Interestingly, these frameworks essentially operate as sequential decision processes. For instance, WireMask-BBO (Shi et al., 2023) initializes the placement order based on the total area of macros within shared nets, while MaskPlace (Lai et al., 2022) employs a heuristic of large/dense first, small/sparse later. This reveals a prevalent bias: these methods predominantly focus on the spatial optimization question of 'Where to place?' while treating the temporal question of 'Who goes first?' as a fixed prior or random permutation. Consequently, the potential of the placement sequence itself remains a dormant, unoptimized dimension.

In a sequential placement process, early decisions define the feasible boundary conditions for all subsequent macros. A single suboptimal placement early in the sequence can trigger a domino effect (Van Leeuwen, 2010), irreversibly constraining the solution space and leading to poor local optima (Ross & Bagnell, 2010; Kahng et al., 2011). To address this sequential dependency, a potential research direction explored by (Majumdar et al., 2024) involves retroactive repair agents—models learned to identify and fix previous placement errors. Similarly, (Deng et al., 2025) employs heuristics to identify and relocate poorly placed macros,

while (Geng et al., 2024) utilizes tree search to mitigate local optima. However, training an agent capable of effectively backtracking and repairing arbitrary mistakes requires impractical exploration and sample complexity for real-world applications (Majumdar et al., 2024). Rather than learning how to repair bad sequences, it is a more elegant and efficient approach to learn how to avoid them by optimizing the input sequence itself.

Despite its critical importance, exploring the impact of sequencing is notoriously difficult for two reasons. First, the signal of ordering is often indirect and masked by the powerful refinement capabilities of complex iterative placement method. Second, evaluating the quality of a single sequence typically requires running a full placement cycle, which is computationally prohibitive for search-based optimization.

To overcome these barriers and unlock the optimization potential of placement sequencing, this paper proposes a novel framework named OrderPlace, which utilizes a low-uncertainty greedy placement strategy as a sensitive probe to magnify the sensitivity of the final placement to the input order. To address the computational cost, OrderPlace introduces a proxy evaluation mechanism. By evaluating macro placement sequences on a simplified proxy task, candidate sequences can be efficiently filtered without incurring the cost of full-scale optimization. Furthermore, the framework leverages LLMs to evolve code-level ordering strategies, discovering generalizable sorting algorithms—ranging from static heuristics to dynamic, physics-inspired rules. In summary, OrderPlace distinguishes itself from previous works through the following key features:

- **Perspective Shift**. We theoretically and empirically demonstrate that the placement sequence is a critical, yet overlooked, dimension of optimization.

- **Methodological Innovation**. We propose the first LLM-driven evolutionary framework specifically designed for macro placement sequencing. By integrating proxy evaluation, we solve the challenge of expensive feedback loops in floorplanning optimization.

- **State-of-the-art (SOTA) Performance**. Our method discovers novel dynamic sorting policies that, when combined with a greedy solver, achieve SOTA results on public benchmarks.

## 2. Related Work

**Macro Placement Sequencing Strategies**. Although macro placement has been extensively studied, strategies for their placement order rely on static heuristic methods or fixed prior processing. GraphPlacement (Mirhoseini et al., 2021) and EfficientPlace (Geng et al., 2024) utilize a larger-first principle, resorting to topological sorting only when sizes are identical. MaskPlace (Lai et al., 2022) and EGPlace (Deng et al., 2025) refine this by prioritizing connectivity degree (dense-first) and then size, while ChiPFormer (Lai et al., 2023) adopts a similar large-first, high-degree-first prior to training its decision transformer. Some approaches derive static scores to guide order. WireMask-BBO (Shi et al., 2023) sorts macro placement order based on the total area of macros within connected nets, and LaMPlace (Geng et al., 2025) calculates a static importance score for each macro. Recent works like Re$^2$MaP (Shi et al., 2025a) and ReMaP (Shi et al., 2025b) introduce more complex priors, such as clustering macros into groups based on corner preferences or ordering based on dataflow intensity (placing weak-flow macros at the periphery first), yet these remain fixed rule-based systems. Methods like DeepPR (Cheng & Yan, 2021) and MaskRegulate (Xue et al., 2024) typically process macros in the raw netlist order or rely on the RL agent to implicitly learn a sequence, without explicitly optimizing the input order as a hyperparameter; similarly, PRNet (Cheng et al., 2022) lacks an explicit sequencing strategy, defaulting to netlist order. Existing works (Liu et al., 2008) predominantly treat placement sequencing as a static preprocessing step governed by fixed heuristics, overlooking its potential as a dynamic, learnable optimization dimension. For more related work about macro placement methods, please refer to Appendix A.

## 3. Preliminaries

**Macro Placement Instance**. A macro placement instance is formally defined as a tuple $I = (M, E, \mathcal{G})$, where:

- $M = \{m_1, m_2, \ldots, m_l\}$ is a set of $l$ macros, where each macro $m_i$ has dimensions $(w_i, h_i)$, area $Ar_i = w_i \times h_i$, and degree (number of connected nets) of macro $m_i$ is $d_i$.

- $E = \{e_1, e_2, \ldots, e_k\}$ is a set of $k$ nets (hyperedges), where each net $e_j \subseteq M$ connects a subset of macros. $\mathcal{N}(m_i)$ is the set of nets that $m_i$ involves. $N(m_i, m_j)$ is the set of nets involves both $m_i$ and $m_j$.

- $\mathcal{G}$ is a discrete placement grid of size $g \times g$.

**Greedy Placement Algorithm**. Given a placement placeing sequence $\Pi = (\pi_1, \pi_2, \ldots, \pi_l)$ which is a permutation of $M$, the greedy placement algorithm $\mathcal{A}$ places macros sequentially as follows:

- **Initialize**: Placed set $P_0 = \emptyset$, Net bounding boxes $\mathcal{B} = \emptyset$.

- **Iterate**: For $t = 1$ to $l$:

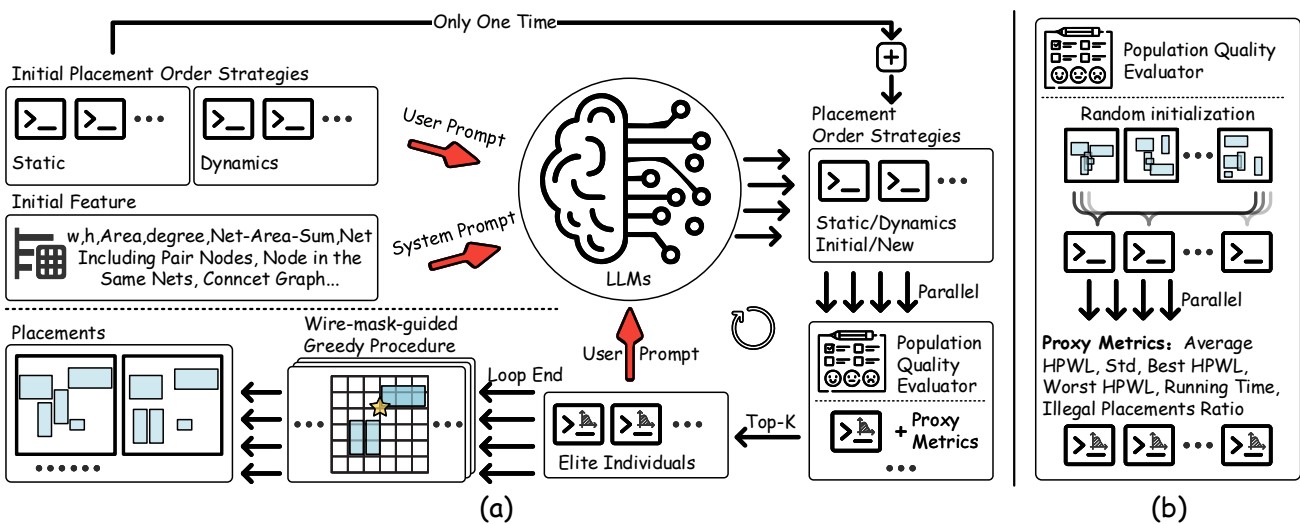

*Figure 1.* Overview of OrderPlace. Part (a) illustrates the overall execution flow of OrderPlace, while part (b) presents the detailed procedure of the Population Quality Evaluator module.

(a) **Select Position**: Determine the optimal position $p_t = (x_t, y_t)$ for the current macro $\pi_t$ that minimizes the incremental wirelength cost:

$$p_t = \underset{p \in \text{Valid}(P_{t-1})}{\text{argmin}} \Delta\text{HPWL}(\pi_t, p, \mathcal{B}) \quad (1)$$

(b) **Update Placement**: $P_t = P_{t-1} \cup \{(\pi_t, p_t)\}$.

(c) **Update Each Net Bounding Boxes**: Recompute $\mathcal{B}$ for all nets connected to $\pi_t$.

• **Return**: The final placement mapping $P_l$.

**Half-Perimeter Wire Length (HPWL)** (Chen et al., 2006). For a given placement $P$, the total HPWL is defined as:

$$\text{HPWL}(P) = \sum_{e_j \in E} \Big[ \Big( \max_{m_i \in e_j} x_i - \min_{m_i \in e_j} x_i \Big) \\ + \Big( \max_{m_i \in e_j} y_i - \min_{m_i \in e_j} y_i \Big) \Big] \quad (2)$$

## 4. Impact of Ordering on Placement Quality

This section formally demonstrates that the sequence in which macros are processed significantly impacts the solution quality of greedy algorithm $\mathcal{A}$. We analyze this utilizing chain and star topologies, which are foundational substructures in very large scale integration (VLSI) netlists.

**Analysis of Chain Topology.**

**Theorem 1**. *For a set of macros connected in a linear chain topology, the ratio of HPWL between the worst-case and best-case ordering is $\Omega(l)$.*

*Proof Sketch.* In an optimal ordering (e.g., sequential), each macro is placed adjacent to its predecessor, resulting in minimal wirelength. In a worst-case ordering (e.g., interleaved), macros are placed without connectivity guidance in the early phases, leading to maximum spatial spread. When intermediate macros are finally placed, they must bridge these distant components, causing the total wirelength to scale quadratically rather than linearly. (See Appendix B.1 for the detailed proof).

**Analysis of Star Topology.**

**Theorem 2**. *For a star topology with a central hub and $l - 1$ spokes placed on a $g \times g$ grid, the expected HPWL of the hub-last ordering is $\Theta(g)$ times worse than the hub-first ordering.*

*Proof Sketch.* If the central hub is placed first, all subsequent spokes cluster tightly around it. Conversely, if the hub is placed last, the spokes—lacking mutual connections—are distributed randomly across the grid. The hub must then connect to these dispersed spokes, making the total wirelength proportional to the grid dimension $g$. (See Appendix B.2 for the detailed proof).

**Summary.** The analyses above demonstrate that for greedy placement algorithms, suboptimal ordering leads to asymptotic degradation. specifically an $\Omega(l)$ degradation for chain topologies and an $\Theta(g)$ degradation for star topologies.

## 5. Methodology

To better explore the optimization potential of placement order for macro placement, we propose an LLM-guided evolutionary search framework for automatic design macro placement order strategies. Our approach consists of three main components: (1) initial placement order strategies, (2)

*Table 1.* The initial macro placement order strategies. $|\cdot|$ denotes the size of the set. $A \to B$ represents sorting first by $A$, then by $B$. $C_t$ represents the placement information that has been completed at time step $t$.

| Strategy Function | Description |
|---|---|
| *Static Ordering Strategies* | |
| $\phi_{\text{area}}(m_i) = -Ar_i$ | ***AreaDesc:*** Prioritize placing larger macros to secure advantageous positions before canvas fragmentation occurs. |
| $\phi_{\text{degree}}(m_i) = -d_i$ | ***DegreeDesc:*** Placing highly connected macros at the forefront enables subsequent macros to achieve optimal positioning relative to critical nodes. |
| $\phi_{\text{area-degree}}(m_i) = (-Ar_i, -d_i)$ | ***Area $\to$ Degree*** |
| $\phi_{\text{degree-area}}(m_i) = (-d_i, -Ar_i)$ | ***Degree $\to$ Area*** |
| $\phi_{\text{net-area}}(m_i) = -\sum_{m_i \in e_j} \sum_{e_j \subseteq m_j} Ar_j$ | – |
| *Dynamic Ordering Strategies* | |
| $\phi_{\text{spring}}(m_i, \mathcal{C}_t) = -\sum_{m_j \in P_t} \kappa_{ij}|\{e \in E : \{m_i, m_j\} \subset e\}|$ | ***Spring Potential:*** Modeling nets as springs connecting macros, this strategy minimizes potential energy, where $\kappa_{ij}$ is proportional to connection strength. |
| $\phi_{\text{field}}(m_i, \mathcal{C}_t) = -\sum_{m_j \in P_t} \alpha|\{e \in E : \{m_i, m_j\} \subset e\}| - \beta d_i$ | ***Field Potential:*** Placed macros generate an attractive field that influences unplaced macros, where $\alpha = 10, \beta = 5$. |
| $\phi_{\text{entropy}}(m_i, \mathcal{C}_t) = -d_i \cdot \frac{c_i^{det(t)}}{c_i^{undet(t)}+1}$ | ***Entropy-based:*** Prioritizes macros that maximize information gain by reducing placement uncertainty ($c_i^{det(t)}$: determined, $c_i^{undet(t)}$: undetermined connections). |
| $\phi_{\text{ham}}(m_i, \mathcal{C}_t) = \frac{\kappa}{|Valid(P_t, m_i)|} + \phi_{\text{spring}}(m_i, C_t)$ | ***Hamiltonian:*** Inspired by classical mechanics, modeling the placement process as a physical system with total energy ($\kappa = 1000$). |

LLM-driven strategy evolution, and (3) wire-mask-guided greedy placement. Figure 1 illustrates the overall framework.

### 5.1. Initial Placement Order Strategies

For LLM-based evolutionary automatic algorithm design, a well-initialized population enables LLMs to better understand task preferences and optimization objectives (Mo et al., 2025). To this end, we manually design a set of macro placement order strategies. Formally, we define a placement order strategy as a priority function:

$$\phi : M \times C \to R_t \quad (3)$$

where $R_t = M \setminus P_t$ is set of remaining (unplaced) macros, assigning a priority score to each macro based on its intrinsic features and the current placement context $C$. The placement order is then determined by sorting macros according to their priority scores in ascending order.

As shown in Table 1, based on how priorities are computed, these strategies are categorized into two classes: static strategies, which compute priorities solely from macro-level geometric and topological features, and dynamic strategies, which adapt priorities according to the evolving placement state, often inspired by physical dynamics simulations. Collectively, these strategies capture diverse characteristics of macros in terms of geometry, topology, and the placement process, providing a diverse and informative initial strategy space for evolutionary search and LLM reasoning.

### 5.2. Design of Placement Order Strategies Driven by LLMs

**Initial Feature.** The macro's width, length, area, the total area of macros contained in the involved nets, as well as the netlist's topological information such as nodes and edges, are all encapsulated in standardized data structures and made accessible to code generated by LLMs through well-defined interfaces.

**LLMs.** OrderPlace supports multiple LLMs backends through a unified adapter interface, including OpenAI GPT-4 (Achiam et al., 2023), Deepseek-V3 (Liu et al., 2024a), and local models via Ollama (Marcondes et al., 2025). The LLM is configured with temperature $\tau = 0.7$ to balance exploration and exploitation in strategy generation (Liu et al., 2024b). For each generation, we query the LLM $Z$ times to produce a batch of candidate strategies, with each query potentially yielding novel combinations or variations of existing approaches.

Besides, all strategies are presented in the form of code. LLM-guided strategy generation employs a two-part prompt structure consisting of a system prompt and a user prompt.

- *System Prompt.* Establishes the LLM's role as a VLSI placement expert, providing (i) problem background on macro ordering impact; (ii) specifications for static and dynamic strategy types; (iii) `PlacementContext` API reference; and (iv) output format requirements.

- *User Prompt.* Dynamically constructed per generation, containing (i) evaluation results of top-$K$ strategies with scores and source code; (ii) identification of the current best strategy; and (iii) guidance for analyzing success factors and exploring novel combinations.

For completeness, we provide the full prompt templates in Appendix C.

**Population Quality Evaluator.** Macro placement is an NP-hard problem, and no deterministic method can directly obtain optimal placement results (Mirhoseini et al., 2021). Typically, a lengthy optimization process is required to achieve satisfactory solutions. Evaluating each placement sequence strategy through complete optimization would incur prohibitive computational costs. To address this challenge, OrderPlace employs a lightweight proxy evaluation mechanism consisting of three stages:

- *Syntax Validation.* The generated code is parsed to verify syntactic correctness and detect potential compilation errors.

- *Functional Testing.* The strategy is executed on a small subset of macros to verify that it produces valid numerical outputs without runtime exceptions.

- *Parallel Monte Carlo Evaluation.* Specifically, for each macro placement sequence strategy, we execute a wire-mask-guided greedy placement process once for each of the $\mathcal{V}$ initial placements (generated using different random seeds), thereby producing $\mathcal{V}$ valid placement solutions. This evaluation is parallelized across $\mathcal{W}$ worker processes to accelerate throughput. A timeout mechanism ($\mathcal{T}$ timeout seconds per strategy) is employed to terminate potentially inefficient or non-terminating strategies.

The fitness score for each strategy is computed as the mean HPWL over successful evaluations:

$$\text{fitness}(\Pi) = \frac{1}{|\mathcal{V}'|} \sum_{i \in \mathcal{V}'} \text{HPWL}(\Pi, \text{seed}_i) \qquad (4)$$

where $\mathcal{V}'$ denotes the set of valid runs excluding timeouts and errors. Additionally, we record proxy metrics including standard deviation, best/worst HPWL, running time, and invalid placement ratio to provide comprehensive strategy characterization.

**Population Evolution.** The evolutionary process maintains a population $\mathcal{Pl}$ of strategies across $Ge$ generations. At each generation $ge$:

- Invalid placement ratio exceeding 50% will be immediately eliminated.

- Select the top-$K$ strategies from $\mathcal{Pl}^{(ge)}$ based on fitness scores.

- Use the selected strategies as context for LLM prompts, generating $Z$ new candidate strategies.

- Evaluate all new strategies using the parallel population quality evaluator.

- Merge new strategies into the population:

$$\mathcal{Pl}^{(ge+1)} = \mathcal{Pl}^{(ge)} \cup Z \qquad (5)$$

- Persist the population state to enable checkpoint recovery.

After $Ge$ generations of LLM-guided evolution, the top-$K'$ strategies undergo fine-tuning via EA optimization guided by the wire-mask-guided greedy procedure in section 5.3. This process yielded the final optimized placement results.

### 5.3. Wire-mask-guided Greedy Procedure

Similar to Wiremask-EA (Shi et al., 2023), given a placement sequence generated by our evolved strategies, Order-Place employs a wire-mask-guided greedy procedure for the actual macro placement. This procedure leverages precomputed wirelength masks to efficiently evaluate candidate positions.

**Wire Mask Construction.** For each macro $m$ to be placed, constructing a wire mask $\mathbf{W}_m \in R^{g \times g}$ over the discretized canvas, where each cell $(i, j)$ stores the estimated HPWL contribution if macro $m$ is placed at that location:

$$\mathbf{W}_m(i, j) = \sum_{e \in \mathcal{N}(m)} \text{HPWL}_e(i, j) \qquad (6)$$

where $\text{HPWL}_e(i, j)$ is the half-perimeter wirelength of net $e$ when macro $m$ is placed at position $(i, j)$, considering the positions of already-placed macros connected to net $e$.

**Greedy Placement.** For each macro in the sequence order:

- Compute the wire mask considering currently placed macros.

- Generate a validity mask $Valid(P_t, m)$ indicating legal positions (no overlaps, within region constraints).

*Table 2.* **HPWL ($\times 10^5$) Achieved by Different Macro Placement Methods on the ISPD2005 Dataset.** The results of baseline methods are taken from EGPlace (Deng et al., 2025) and BBOPlace-Bench (Xue et al., 2025). All results, except those of the deterministic method NTUPlace3, are averaged over 5 runs with different random seeds and reported as mean ± std. Symbols '+', '−', and '≈' indicate the number of circuits where the method performs significantly better than, worse than, or comparable to OrderPlace, based on the Wilcoxon rank-sum test at a 0.05 significance level. The best results are marked in **bold**.

| Method | adaptec1 | adaptec2 | adaptec3 | adaptec4 | bigblue1 | bigblue3 | +/−/≈ | Avg. Rank |
|---|---|---|---|---|---|---|---|---|
| SP-SA | $18.84 \pm 4.62$ | $117.36 \pm 8.73$ | $115.48 \pm 7.56$ | $120.03 \pm 4.25$ | $5.12 \pm 1.43$ | $164.70 \pm 19.55$ | 0/6/0 | 8.67 |
| NTUPlace3 | 26.62 | 321.17 | 328.44 | 462.93 | 22.85 | 455.53 | 0/6/0 | 11.17 |
| RePlace | $16.19 \pm 2.10$ | $153.26 \pm 29.01$ | $111.21 \pm 11.69$ | $37.64 \pm 1.05$ | $2.45 \pm 0.06$ | $119.84 \pm 34.43$ | 0/6/0 | 6.83 |
| DreamPlace | $15.81 \pm 1.64$ | $140.79 \pm 26.73$ | $121.94 \pm 25.05$ | $37.41 \pm 0.87$ | $2.44 \pm 0.06$ | $107.19 \pm 29.91$ | 0/6/0 | 6.33 |
| GraphPlace | $30.10 \pm 2.98$ | $351.71 \pm 38.20$ | $358.18 \pm 13.95$ | $151.42 \pm 9.72$ | $10.58 \pm 1.29$ | $357.48 \pm 47.83$ | 0/6/0 | 10.83 |
| DeepPR | $19.91 \pm 2.13$ | $203.51 \pm 6.27$ | $347.16 \pm 4.32$ | $311.86 \pm 56.74$ | $23.33 \pm 3.65$ | $430.48 \pm 12.18$ | 0/6/0 | 11.00 |
| MaskPlace | $7.62 \pm 0.67$ | $75.16 \pm 4.97$ | $100.24 \pm 13.54$ | $87.99 \pm 3.25$ | $3.04 \pm 0.06$ | $90.04 \pm 4.83$ | 0/6/0 | 6.17 |
| Chipformer | $6.62 \pm 0.05$ | $67.10 \pm 5.46$ | $76.70 \pm 1.15$ | $68.80 \pm 1.59$ | $2.95 \pm 0.04$ | $72.92 \pm 2.56$ | 0/6/0 | 5.17 |
| EfficientPlace | $5.94 \pm 0.04$ | $46.79 \pm 1.60$ | $56.35 \pm 0.99$ | $58.47 \pm 1.61$ | $2.14 \pm 0.01$ | $58.38 \pm 0.54$ | 0/6/0 | 2.67 |
| WireMask-EA | $6.15 \pm 0.05$ | $64.38 \pm 4.43$ | $58.18 \pm 1.04$ | $59.52 \pm 1.71$ | $2.15 \pm 0.01$ | $59.85 \pm 3.39$ | 0/6/0 | 3.67 |
| EGPlace | $\mathbf{5.72 \pm 0.01}$ | $37.69 \pm 1.08$ | $60.13 \pm 1.83$ | $56.08 \pm 0.43$ | $2.20 \pm 0.01$ | $52.41 \pm 8.16$ | 1/5/0 | 2.50 |
| **OrderPlace** | $5.75 \pm 0.06$ | $\mathbf{30.72 \pm 1.47}$ | $\mathbf{54.82 \pm 0.38}$ | $\mathbf{49.88 \pm 0.28}$ | $\mathbf{2.00 \pm 0.00}$ | $\mathbf{36.72 \pm 0.44}$ | | **1.17** |

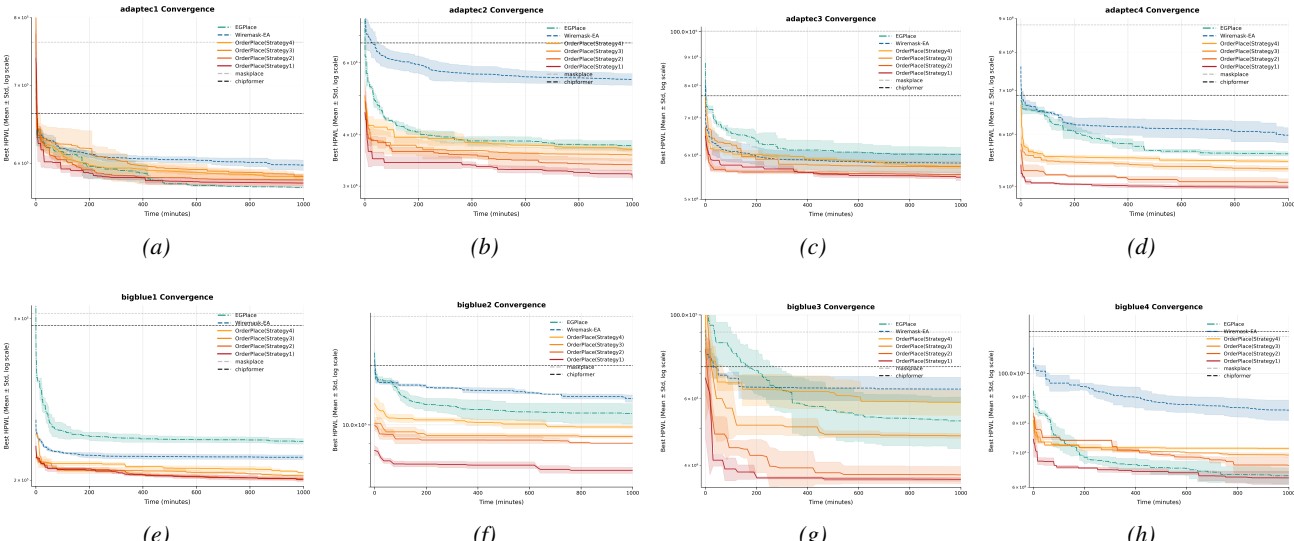

|     |     |     |     |
|-----|-----|-----|-----|
| *(a)* | *(b)* | *(c)* | *(d)* |
| *(e)* | *(f)* | *(g)* | *(h)* |

*Figure 2.* Comparison of HPWL Trend Over the Runtime(s). In these trajectories, OrderPlace(Strategy$Number$) denotes the placement process utilizing the $Number$-th best ordering strategy identified by our framework. For each dataset, we select the top 4 strategies and perform parallel macro placement optimization, resulting in a total of 24 strategies. For specific strategy details, refer to Appendix F.

- Select the position minimizing the masked wirelength:

$$(\hat{i}, \hat{j}) = \arg\min_{(i,j)} \mathbf{W}_m(i,j) \cdot Valid(P_t, m) \quad (7)$$

- Place the macro and update the canvas state.

This greedy procedure ensures that each macro is placed at a locally optimal position given the current canvas state, while the evolved sequence strategy determines the global ordering that influences the overall placement quality.

## 6. Experiments

**Benchmarks and Settings.** Following prior macro placement methods (Shi et al., 2023; Deng et al., 2025; Geng et al., 2024), we validate the effectiveness of OrderPlace on

the ISPD 2005 benchmark suite (Nam et al., 2005). The ISPD 2005 suite contains eight chip designs; details of these circuits are provided in Appendix D. We compare OrderPlace against leading placement methods, including the simulated-annealing based SP-SA (Murata et al., 2002); analytic and gradient-based placers NTUPlace3 (Chen et al., 2008), RePlace (Cheng et al., 2018), DreamPlace (Lin et al., 2019); RL approaches GraphPlace (Mirhoseini et al., 2021), DeepPR (Cheng & Yan, 2021), MaskPlace (Lai et al., 2022), Chipformer (Lai et al., 2023), and EfficientPlace (Geng et al., 2024); and hybrid-optimization WireMask-EA (Shi et al., 2023), EGPlace (Deng et al., 2025). All experiments of OrderPlace are conducted on a machine equipped with eight NVIDIA RTX A6000 GPUs and four Intel(R) Xeon(R) Platinum 8374C CPUs running at 2.70 GHz. Both the population quality evaluator and the final

*Table 3.* Summary of Mathematical Components in LLM-Generated Strategies, where $\rho = |P_t|/|M|$ is placement progress ratio, $u_e = |e| - |e \cap P_t| - 1$ is macros in net $e$ excluding current candidate. $Var$ and $scale$ are the variance and normalization functions, respectively.

| Component | Typical Form | Frequency |
|---|---|---|
| Connection Strength | $\sum |N(m_i, m_j)|^{\alpha}, \alpha \in [1.0, 1.5]$ | 100% (24/24) |
| Degree Factor | $d_i, \ln(d_i + 1), \text{ or } \sqrt{d_i}$ | 100% (24/24) |
| Phase-Adaptive Weights | $weight(\rho)$ piecewise function | 87.5% (21/24) |
| Net Criticality | $1/\sqrt{|e|}$ or $\exp(-|e|/k)$ | 87.5% (21/24) |
| Entropy Reduction | $c^{\text{det}}/f(c^{\text{undet}})$ | 75% (18/24) |
| Area Factor | $\sqrt{Ar_i}$ or $\ln(Ar_i + 1)$ | 75% (18/24) |
| Net Closure Bonus | High reward if $u_e = 0$ | 62.5% (15/24) |
| Spatial Compactness | $1/(1 + \sqrt{Var}/scale)$ | 50% (12/24) |

wire-mask-guided greedy procedure evaluate order strategies in parallel using eight threads. The LLM-driven order strategy generation is run for 4 iterations by default, with the LLM generating $Z = 4$ candidate strategies in parallel per iteration. Further experimental details are available in Appendix D. The source code and all macro placement order strategies are publicly available at `https://github.com/Explorermomo/OrderPlace`.

## 6.1. Macro Placement Results

We evaluate the macro placement performance on the standard ISPD2005 benchmarks, with detailed HPWL comparisons summarized in Table 2. OrderPlace demonstrates superior placement quality, achieving the best (lowest) HPWL values on 7 out of 8 circuits—adaptec2∼4, and bigblue1∼4—and securing the top average rank of 1.17 among all twelve competing methods (the result of bigblue2 and bigblue4 cases is provided in Appendix E). In a direct comparison with the strongest baseline, EGPlace, OrderPlace performs significantly better on 7 circuits based on the Wilcoxon rank-sum test at a 0.05 significance level. Beyond achieving superior final solution quality, OrderPlace exhibits remarkable computational efficiency; as illustrated in Figure 2, our method converges significantly faster than EGPlace, which is renowned for its rapid convergence capabilities. Notably, OrderPlace(Strategy1) consistently demonstrates the steepest descent in HPWL, navigating the solution space to reach lower convergence points much earlier than both EGPlace and WireMask-EA across all benchmarks (Figure 2a-h). This confirms that proactive optimization of the placement sequence not only unlocks better local optima but also fundamentally accelerates the greedy solver's convergence toward high-quality global solutions.

## 6.2. Analysis of LLM-Generated Strategies

We analyze the placement sequence strategies discovered through our LLM-guided evolutionary search framework across eight ISPD benchmark circuits. Table 8 in Appendix F summarizes the top-4 strategies for each dataset.

**Overview of Discovered Strategies.** Our experimental results reveal several key findings:

- **Dominance of LLM-generated strategies**: Out of 32 top-4 positions across all datasets, 24 (75%) are occupied by LLM-generated strategies, while only 8 positions are held by built-in heuristics.

- **Dynamic strategies prevail**: All 8 top-1 positions are held by LLM-generated dynamic strategies that adapt their behavior based on placement progress.

- **No static strategies in top positions**: None of the top-1 strategies across any dataset employ static ordering, demonstrating the importance of adaptive decision-making.

**Taxonomy of Discovered Strategy Patterns.** As shown in Table 3, through analysis of the 24 LLM-generated strategies, we identify some recurring design components that emerge across different datasets. Next, several patterns formed by these components are analyzed.

*Pattern 1: Gravitational/Field Models.* These strategies model placed macros as mass points generating attractive fields. The gravitational force on an unplaced macro $m_i$ is computed as:

$$S_g(m_i) = \sum_{m_j \in P_t} |N(m_i, m_j)|^{\alpha} \cdot \psi(m_i, m_j) \quad (8)$$

where $\alpha \in [1.0, 1.5]$ controls the superlinear scaling of connection strength, and $\psi(\cdot)$ represents optional mass factors (e.g., $\sqrt{Ar_i \cdot Ar_j}$ for area-weighted gravity).

*Pattern 2: Entropy Reduction Models.* Inspired by information theory, these strategies prioritize macros that maximize information gain by reducing placement uncertainty:

$$S_{\text{entropy}}(m_i) = \frac{c_i^{\text{det}}}{f_1(c_i^{\text{undet}})} \cdot f_2(d_i) \quad (9)$$

where $c_i^{\text{det}}$ and $c_i^{\text{undet}}$ denote determined and undetermined connections, $f_1(\cdot)$ is typically $\sqrt{\cdot + 1}$ or $(\cdot + 1)$, and $f_2(d_i)$ is a degree-dependent factor.

*Pattern 3: Net Closure Prioritization.* These strategies explicitly prioritize completing nearly-finished nets:

$$S_{\text{closure}}(e) = \begin{cases} \text{High\_Bonus} \cdot f(e) & \text{if } |e \setminus P_t| = 1 \\ \left(\frac{|e \cap P_t|}{|e|}\right)^{\beta} \cdot f(e) & \text{otherwise} \end{cases} \quad (10)$$

where $f(e) = \exp(-|e|/k)$ or $1/\sqrt{|e|}$ represents net criticality, High\_Bonus denotes a fixed reward obtained upon the completion of macro placement in the current net $e$, and $\beta \geq 2$ provides superlinear scaling for high completion ratios.

*Table 4.* Comparison on HPWL($\times 10^5$) and Congestion Results. We standardize the RUDY value of OrderPlace(Strategy1) to 1.00. The best results are marked in **bold**.

| Benchmarks | adaptec1 | | adaptec2 | | adaptec3 | | adaptec4 | | bigblue1 | | bigblue2 | | bigblue3 | | bigblue4 | |
| Metrics | HPWL | Cong. | HPWL | Cong. | HPWL | Cong. | HPWL | Cong. | HPWL | Cong. | HPWL | Cong. | HPWL | Cong. | HPWL | Cong. |
|---|---|---|---|---|---|---|---|---|---|---|---|---|---|---|---|---|
| WireMask-EA | 5.89 | 2.02 | 52.63 | 2.07 | 54.72 | 1.41 | 57.38 | 1.28 | 2.10 | 1.64 | 11.06 | 1.28 | 62.69 | 1.02 | 76.12 | 1.39 |
| EGPlace | **5.68** | 1,61 | 37.99 | 1.64 | 63.05 | 1.41 | 56.09 | 1.27 | 2.23 | 1.62 | 10.49 | 1.26 | 50.50 | 1.65 | **58.96** | **0.75** |
| OrderPlace (Strategy4) | 5.80 | 1.59 | 33.54 | **0.95** | 55.55 | 1.45 | 53.46 | 1.40 | 2.00 | 1.02 | 9.54 | 1.19 | 52.25 | 1.00 | 71.02 | 1.00 |
| OrderPlace (Strategy3) | 5.77 | 1.83 | 33.33 | 1.04 | 54.36 | 1.44 | 51.88 | 1.36 | 1.99 | 1.01 | 9.07 | 1.15 | 46.52 | 0.98 | 64.69 | 1.02 |
| OrderPlace (Strategy2) | 5.70 | 1.71 | 30.63 | 1.15 | **52.81** | 1.15 | 48.38 | 1.17 | **1.97** | 1.38 | 8.78 | 1.09 | **35.22** | 1.02 | 61.82 | 1.02 |
| OrderPlace (Strategy1) | **5.68** | **1.00** | 28.66 | 1.00 | 52.85 | **1.00** | **48.31** | **1.00** | 2.00 | **1.00** | 7.38 | **1.00** | 35.24 | **1.00** | 60.57 | 1.00 |

*Table 5.* Comparisons of HPWL ($\times 10^7$) on mixed-size placement task. The results of the comparison methods are taken from the BBOPlace-Bench (Xue et al., 2025) and EGPlace (Deng et al., 2025) paper The best results are highlighted in bold.

| Method | adaptec1 | adaptec2 | adaptec3 | adaptec4 | bigblue1 | bigblue3 | Avg. Rank |
|---|---|---|---|---|---|---|---|
| DreamPlace | $9.62 \pm 0.78$ | $12.45 \pm 3.31$ | $17.18 \pm 0.51$ | $40.04 \pm 3.87$ | $8.30 \pm 0.06$ | $38.22 \pm 1.48$ | 4.33 |
| MaskPlace + DreamPlace | $10.86 \pm 0.18$ | $12.98 \pm 0.58$ | $26.14 \pm 0.07$ | $26.14 \pm 0.07$ | $10.64 \pm 0.01$ | $54.98 \pm 1.06$ | 5.83 |
| WireMask-EA + DreamPlace | $8.93 \pm 0.01$ | $9.20 \pm 0.05$ | $21.72 \pm 0.01$ | $20.51 \pm 0.01$ | $10.35 \pm 0.02$ | $42.52 \pm 0.11$ | 4.25 |
| EfficientPlace + DreamPlace | $\mathbf{7.20 \pm 0.12}$ | $9.20 \pm 0.61$ | $16.49 \pm 1.07$ | $14.70 \pm 0.25$ | $8.67 \pm 0.10$ | $28.48 \pm 0.96$ | 2.25 |
| EGPlace + DreamPlace | $7.53 \pm 0.11$ | $\mathbf{9.06 \pm 0.36}$ | $\mathbf{14.15 \pm 0.19}$ | $15.69 \pm 0.09$ | $8.99 \pm 0.06$ | $29.08 \pm 0.23$ | 2.33 |
| OrderPlace + DreamPlace | $8.11 \pm 0.06$ | $11.92 \pm 0.44$ | $14.87 \pm 0.36$ | $\mathbf{14.28 \pm 0.27}$ | $\mathbf{8.29 \pm 0.07}$ | $\mathbf{27.96 \pm 0.13}$ | **2.00** |

**Case Study on the OrderPlace-Strategy1 of Adaptec1.**

The OrderPlace-Strategy1 (`gravity_entropy`) of adaptec1 combines gravitational attraction from placed macros with information-theoretic metrics for net closure.

*Initial Placement ($P_t = \emptyset$).*

$$\phi(m_i) = -\sqrt{d_i \cdot Ar_i} \cdot 1000 \quad (11)$$

This prioritizes macros with high combined degree-area product, establishing a well-connected foundation.

*Dynamic Placement.*

$$\phi(m_i, \mathcal{C}_t) = -\Big( 0.3 \cdot S_{grvity_i} + 0.2 \cdot S_{pnet_i} \\ + 0.4 \cdot S_{cnet_i} + 0.1 \cdot \sqrt{d_i \cdot Ar_i} \Big) \quad (12)$$

where

$$\begin{cases} S_{grvity_i} = \sum_{m_j \in P_t} |N(m_i, m_j)|^{1.5} \\ S_{pnet_i} = \sum_{\substack{e \in \mathcal{N}(m_i): \\ |e \cap P_t| > 0}} 4 \cdot \frac{|e \cap P_t|}{|e|} \cdot (1 - \frac{|e \cap P_t|}{|e|}) \\ S_{cnet_i} = \sum_{e \in \mathcal{N}(m_i): u_e = 0} 10 \ln(|e| + 1) \\ \quad + \sum_{e \in \mathcal{N}(m_i): u_e \leq 2} 5 \ln(|e| + 1) \end{cases} \quad (13)$$

The overall scoring function is composed of three complementary components. The $S_{grvity}$ captures the attraction induced by already placed macros, using a superlinear scaling to emphasize the influence of larger or more connected macros. The $S_{pnet}$ quantifies the placement progress of partially placed nets through a bell-shaped weighting scheme, encouraging balanced advancement without premature convergence. Finally, the $S_{cnet}$ provides explicit rewards for nets that are fully completed or close to completion, thereby promoting net closure and improving global connectivity during the placement process.

Overall, these results further confirm that adaptive, multifactor strategies are critical for effective macro placement, **with additional details and examples provided in Appendix F.**

### 6.3. Additional Results

**Congestion Results.** The observed results of Table 4 align with the findings of WireMask-EA (Shi et al., 2023)—lower HPWL may help reduce congestion. This correlation is particularly evident in the bigblue4 benchmark, where EGPlace achieves both the lowest HPWL and the minimum congestion. Overall, OrderPlace (Strategy 1) demonstrates the most consistent routability, securing the optimal standardized congestion score of 1.00 in six out of the eight benchmarks.

**Mixed-size Placement Results.** We adopted the same two-stage global placement framework as EfficientPlace (Geng et al., 2024) and EGPlace (Deng et al., 2025). As shown in Table 5, compared with EGPlace and EfficientPlacement, OrderPlace achieves a leading performance on half of the six benchmarks and attains the best average ranking. In particular, the comparison with Wiremask-EA demonstrates that the placement order of macros is also an important feature dimension. More experimental results can be found in Appendix G.

# 7. Conclusion and Future Work

This work presents the first systematic evidence that placement sequencing, traditionally overlooked as a static pre-processing step, serves as a critical determinant in placement optimization. We propose OrderPlace, an LLM-driven framework that automates ordering strategy discovery via proxy evaluation. Extensive experimental results demonstrate that the strategies discovered by OrderPlace effectively mitigate suboptimality in sequence-decision-based placement, particularly during early-stage design.

**Future Work and Limitation.** This study is limited to evaluating the impact of placement order on mask-guided greedy methods. The interaction with stochastic, learning-based methods remains unexplored. Future work will explore how placement ordering enhances deep learning-driven methods.

# Acknowledgments

This work was supported in part by the National Natural Science Foundation of China under Grant 62471371, and in part by the Fundamental Research Funds for the Central Universities under Grant YJSJ26008.

# Impact Statement

This paper presents work whose goal is to advance the field of machine learning. There are many potential societal consequences of our work, none of which we feel must be specifically highlighted here.

We provide the following information to ensure the reproducibility of our proposed OrderPlace. Implementation details are given in Appendix D. All strategy details are provided in Appendix F.

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

# A. More Related Work

**Macro Placement Methods**. Existing macro placement methodologies can be broadly categorized into analytical, RL-based, and constructive approaches. Analytical methods, such as ePlace (Lu et al., 2015) and DreamPlace (Lin et al., 2019), formulate placement as a continuous optimization problem, leveraging gradient descent to minimize wirelength and density penalties. While efficient, these methods often struggle with the discrete nature of macro non-overlap constraints and non-differentiable objectives. RL-based methods, pioneered by AlphaChip (Mirhoseini et al., 2021) and followed by MaskPlace (Lai et al., 2022) and ChiPFormer (Lai et al., 2023), model placement as a sequential decision process (Alagoz et al., 2010). While effective, they often require substantial computational resources for training and can suffer from sample inefficiency. Constructive and Hybrid methods, such as WireMask-EA (Shi et al., 2023), combine global search algorithms (e.g., Evolutionary Algorithms(Cohoon & Paris, 1987)) with deterministic greedy solvers. WireMask-EA specifically utilizes a wiremask-guided greedy insertion strategy to rapidly generate valid placements. Unlike analytical methods that optimize all coordinates simultaneously, or RL methods that entangle position and order in complex policies, OrderPlace specifically employs the deterministic greedy strategy as a sensitive probe to rigorously isolate and quantify the impact of input sequencing, unclouded by the stochastic noise of gradient descent or annealing processes.

# B. Deferred Proofs

### B.1. Proof of Theorem 1 (Chain Topology)

**Theorem 1**. *For a set of macros connected in a linear chain topology, the ratio of HPWL between the worst-case and best-case ordering is $\Omega(l)$.*

*Proof.* Consider a set of $l$ macros $M$ connected by nets $E = \{(m_i, m_{i+1}) \mid 1 \leq i < l\}$.

1. **Best-Case Ordering**: Let $\Pi^* = (m_1, m_2, \ldots, m_l)$. The algorithm $\mathcal{A}$ places $m_1$ at an initial position. For every subsequent step $t > 1$, macro $m_t$ is connected to the previously placed $m_{t-1}$ via a 2-pin net. To minimize $\Delta$HPWL, $\mathcal{A}$ places $m_t$ adjacent to $m_{t-1}$. Consequently, the Manhattan distance for each net is exactly 1.

$$\text{HPWL}(\mathcal{A}(\Pi^*)) = \sum_{i=1}^{l-1} 1 = l - 1 \tag{14}$$

2. **Worst-Case Ordering**: Let $\Pi_{\text{worst}} = (m_1, m_3, \ldots, m_{2\lceil l/2 \rceil - 1}, m_2, m_4, \ldots)$. In the first phase, $\mathcal{A}$ places all odd-indexed macros. Since there are no edges between $m_i$ and $m_j$ where both $i, j$ are odd, these macros share no active connections during this phase. Lacking connectivity guidance, a greedy heuristic (or random tie-breaking) may maximize spatial spread, placing macros at extrema of the grid (e.g., corners or boundaries).

   In the second phase, $\mathcal{A}$ places the even-indexed macros. Each $m_{2i}$ connects to $m_{2i-1}$ and $m_{2i+1}$. If the odd macros are dispersed such that the distance between $m_{2i-1}$ and $m_{2i+1}$ is proportional to the grid dimension $g$ (where usually $g \propto l$ or $g \propto \sqrt{l}$), the forced placement of $m_{2i}$ results in a wirelength proportional to the distance between its neighbors. In a pathological case where the odd macros are maximally separated, the total wirelength approaches $\Omega(l^2)$ (assuming $g \approx l$).

The variation ratio is derived as:
$$\frac{\max_\Pi \text{HPWL}(\mathcal{A}(\Pi))}{\min_\Pi \text{HPWL}(\mathcal{A}(\Pi))} = \frac{\Omega(l^2)}{O(l)} = \Omega(l) \tag{15}$$

Thus, poor ordering can degrade quality linearly with respect to the problem size. $\square$

### B.2. Proof of Theorem 2 (Star Topology)

**Theorem 2**. *For a star topology with a central hub and $l - 1$ spokes placed on a $g \times g$ grid, the expected HPWL of the hub-last ordering is $\Theta(g)$ times worse than the hub-first ordering.*

*Proof.* Let $M$ consist of a hub $H$ and spokes $S = \{s_1, \ldots, s_{l-1}\}$, with nets $E = \{(H, s_i) \mid 1 \leq i < l\}$.

1. **Hub-First Ordering**: $\Pi_1 = (H, s_1, \ldots, s_{l-1})$. $\mathcal{A}$ places $H$ first (e.g., at the center $(\frac{g}{2}, \frac{g}{2})$). Subsequently, each spoke $s_i$ has an active connection to $H$. The greedy choice places each $s_i$ adjacent to $H$.

$$\text{HPWL}(\mathcal{A}(\Pi_1)) \approx l - 1 \tag{16}$$

2. **Hub-Last Ordering**: $\Pi_2 = (s_1, \ldots, s_{l-1}, H)$. During the placement of spokes, there are no mutual connections between any $s_i, s_j$. The algorithm receives no guidance ($\Delta\text{HPWL} = 0$). Assuming a uniform distribution for tie-breaking over the grid domain $[0, g]$, the spokes are effectively randomly distributed.

   The expected position of any spoke is $E[x_i] = g/2$. When $H$ is finally placed, it connects to all spokes. Even if $H$ is placed at the centroid to minimize total displacement, the expected HPWL is dominated by the spread of the spokes. Approximating the discrete sum with a continuous integral, the expected Manhattan distance between the hub and a random spoke is:

$$E[|x_H - x_i| + |y_H - y_i|] \approx 2 \cdot E[|X - g/2|] = 2 \int_0^g \left| x - \frac{g}{2} \right| \frac{1}{g} dx = \frac{g}{2} \tag{17}$$

   Summing over $l - 1$ connections (approximating $l - 1 \approx l$ for large $l$):

$$E[\text{HPWL}(\mathcal{A}(\Pi_2))] \approx l \cdot \frac{g}{2} \tag{18}$$

The degradation ratio is:

$$\frac{E[\text{HPWL}(\mathcal{A}(\Pi_2))]}{\text{HPWL}(\mathcal{A}(\Pi_1))} \approx \frac{l \cdot (g/2)}{l} = \frac{g}{2} \tag{19}$$

This confirms that ordering impacts solution quality by a factor proportional to the grid dimension $g$. $\qquad\square$

## C. Complete Prompt Templates

This appendix provides the complete prompt templates used in our LLM-guided evolutionary search framework. The prompts are designed to guide the LLM in generating novel macro placement sequence strategies.

### C.1. System Prompt

The system prompt establishes the LLM's role and provides comprehensive technical context:

*Listing 1.* System Prompt Template

```
You are an expert in VLSI macro placement optimization. Your task is to evolve and improve macro placement ordering
    strategies through creative mutations and combinations.

## Background
In macro placement, the ORDER in which macros are placed significantly affects the final HPWL (Half-Perimeter Wire
    Length). We use a greedy placer that places one macro at a time.

## Strategy Types
There are TWO types of ordering strategies:

### 1. Static Strategy (is_static = True)
- Generates a complete ordering at once based on macro attributes
- Implements 'generate_order(self, ctx)' method that returns a sorted list of node IDs
- Example: Sort all macros by area, degree, or combined metrics

### 2. Dynamic Strategy (is_static = False)
- Selects the next macro to place based on current placement state
- Implements 'compute_priority(self, node_id, ctx)' and 'select_next(self, ctx)' methods
- Can adapt to placement progress (e.g., connectivity to already placed macros)

## Available APIs

### ctx: PlacementContext
- 'ctx.get_node_attr(node_id, 'area', default=0)' - Get macro area
- 'ctx.get_node_attr(node_id, 'degree', default=0)' - Get connectivity degree
- 'ctx.get_node_attr(node_id, 'x', default=0)' - Get macro width
- 'ctx.get_node_attr(node_id, 'y', default=0)' - Get macro height
- 'ctx.node_info' - Dict of all macro info {node_id: {area, degree, x, y, ...}}
```

```
- `ctx.remaining_macros` - Set of unplaced macro IDs
- `ctx.placed_macros` - Dict of placed macros {node_id: {loc_x, loc_y, ...}}
- `ctx.get_shared_nets(node_a, node_b)` - Get list of shared nets between two macros
- `ctx.node_nets.get(node_id, [])` - Get list of nets connected to a macro
- `ctx.net_info` - Dict of net info {net_id: {nodes: {...}}}
- `ctx.hpwl_info` - Current bounding boxes for each net
- `ctx.grid_num`, `ctx.grid_size` - Grid parameters

### Base Classes
- Static strategies inherit from `OrderGenerator`
- Dynamic strategies inherit from `DynamicOrderGenerator`

### Available imports
- math (math.sqrt, math.log, math.exp, math.ceil, etc.)
- random
- numpy as np

## Requirements
- Return a COMPLETE class definition
- For static: set `is_static = True` and implement `generate_order()`
- For dynamic: set `is_static = False` and implement `compute_priority()` + `select_next()`
- Use <CODE_SNIPPET></CODE_SNIPPET> to wrap the class code
- Be creative and try novel combinations!
```

## C.2. User Prompt

The user prompt is dynamically constructed at each generation, providing dataset-specific context and top-performing strategies:

*Listing 2.* User Prompt Template

```
Based on the evaluation results for dataset '{dataset}', here are the TOP {top_k} performing strategies:

{top_strategies}

The BEST strategy achieved min_hpwl={best_min}, mean_hpwl={best_mean}

Your task: Generate a NEW strategy class that could potentially outperform these.
Consider:
1. What made the best strategies successful?
2. Are there unexplored combinations of factors?
3. Could dynamic adaptation (based on placement progress) help?
4. Would a static or dynamic approach work better?

Respond with a COMPLETE Python class definition:

For STATIC strategy:
<CODE_SNIPPET>
class NewStrategy(OrderGenerator):
    name = "new_strategy"
    description = "Description_of_your_strategy"
    is_static = True

    def generate_order(self, ctx: PlacementContext) -> List[str]:
        nodes = list(ctx.remaining_macros) if ctx.remaining_macros else list(ctx.node_info.keys())
        # Your sorting logic here
        return sorted(nodes, key=lambda x: your_key_function(x), reverse=True)
</CODE_SNIPPET>

For DYNAMIC strategy:
<CODE_SNIPPET>
class NewStrategy(DynamicOrderGenerator):
    name = "new_strategy"
    description = "Description_of_your_strategy"
    is_static = False

    def compute_priority(self, node_id: str, ctx: PlacementContext) -> float:
        # Your priority logic here (lower = higher priority)
        return score

    def select_next(self, ctx: PlacementContext) -> Optional[str]:
        if not ctx.remaining_macros:
            return None
        return min(ctx.remaining_macros, key=lambda x: self.compute_priority(x, ctx))
</CODE_SNIPPET>
```

## C.3. Top Strategies Format

The {top_strategies} placeholder in the user prompt is populated with detailed information about each top-performing strategy:

*Listing 3.* Top Strategies Format

```
### Strategy 1: {strategy_name}
- min_hpwl: {min_hpwl}
- mean_hpwl: {mean_hpwl}
- valid_ratio: {valid_ratio}
- Description: {description}
- Code:
```python
{source_code}
```

### Strategy 2: {strategy_name}
...
```

## C.4. Example: Populated User Prompt

The following shows an example of a fully populated user prompt during evolution:

*Listing 4.* Example Populated User Prompt

```
Based on the evaluation results for dataset 'adaptec1', here are the TOP 3 performing strategies:

### Strategy 1: net_area_desc
- min_hpwl: 8,234,567
- mean_hpwl: 8,456,789
- valid_ratio: 100.0%
- Description: Sort macros by degree*area in descending order
- Code:
```python
def generate_order(self, ctx: PlacementContext) -> List[str]:
    nodes = list(ctx.node_info.keys())
    return sorted(nodes,
        key=lambda x: ctx.get_node_attr(x, 'degree', 0) * ctx.get_node_attr(x, 'area', 0),
        reverse=True)
```

### Strategy 2: spring_potential
- min_hpwl: 8,345,678
- mean_hpwl: 8,567,890
- valid_ratio: 98.5%
- Description: Dynamic strategy using spring potential model
- Code:
```python
def compute_priority(self, node_id: str, ctx: PlacementContext) -> float:
    if not ctx.placed_macros:
        return -ctx.get_node_attr(node_id, 'degree', 0)
    connectivity = sum(len(ctx.get_shared_nets(node_id, pid))
                    for pid in ctx.placed_macros)
    return -(connectivity * 10 + ctx.get_node_attr(node_id, 'degree', 0))
```

### Strategy 3: area_desc
- min_hpwl: 8,456,789
- mean_hpwl: 8,678,901
- valid_ratio: 100.0%
- Description: Sort macros by area in descending order (largest first)
- Code:
```python
def generate_order(self, ctx: PlacementContext) -> List[str]:
    nodes = list(ctx.node_info.keys())
    return sorted(nodes, key=lambda x: ctx.get_node_attr(x, 'area', 0), reverse=True)
```

The BEST strategy achieved min_hpwl=8,234,567, mean_hpwl=8,456,789

Your task: Generate a NEW strategy class that could potentially outperform these.
...
```

## D. Dataset & Experimental Details

Table 6 provides detailed statistics of the eight circuits from the ISPD 2005 benchmark, which are used as our test dataset. The "Place Number" column specifies the number of macros selected for placement in our study. For bigblue2 and bigblue4, due to the large number of macros, we follow the setting of EGPlace (Deng et al., 2025) and select 1024 macros for a fair comparison.

*Table 6.* Statistics of public benchmark circuits.

| Circuit | Macros | Place Number | Hard Macros | Standard Cells | Nets | Pins | Area Util (%) |
|---------|--------|--------------|-------------|----------------|------|------|---------------|
| adaptec1 | 543 | 543 | 63 | 210904 | 221142 | 944063 | 55.62 |
| adaptec2 | 566 | 566 | 159 | 254457 | 266009 | 1069482 | 74.46 |
| adaptec3 | 723 | 723 | 201 | 450927 | 466758 | 1875039 | 61.51 |
| adaptec4 | 1329 | 1329 | 92 | 494716 | 515951 | 1912420 | 48.62 |
| bigblue1 | 560 | 560 | 32 | 277604 | 284479 | 1144691 | 31.58 |
| bigblue2 | 23084 | 1024 | 52 | 534782 | 577235 | 2122282 | 32.43 |
| bigblue3 | 1293 | 1293 | 138 | 1095519 | 1123170 | 3833218 | 66.81 |
| bigblue4 | 8170 | 1024 | 52 | 2169183 | 2229886 | 8900078 | 35.68 |

**More Details of Experimental Setting.**

For the Population Quality Evaluator, the evaluation time for each ordering strategy is limited to 1800 seconds, and the number of Monte Carlo samples is set to 50 by default. We choose claude-sonnet-4.5 as the LLM. In each user prompt, the top-4 elite strategies are provided to the LLM, which helps guide it to generate ordering strategies that better match the preferences of the dataset. After the LLM-based strategy design loop is completed, the top-4 ordering strategies are again selected and executed in parallel using the Wire-mask-guided Greedy Procedure. The placement with the minimum HPWL among them is taken as the final placement result. For the specification of the discrete placement grid, we use a resolution of $224 \times 224$ by default.

For each run of OrderPlace, the time limit is set to 1000 minutes. Meanwhile, for a fair comparison, we run the official codebases of EGPlace and EfficientPlace and report their best results achieved within the same 1000-minutes time limit.

## E. Results on the "bigblue2" and "bigblue4" Circuits

In this section, we provide a detailed analysis of the performance on "bigblue2" and "bigblue4," which represent significantly more challenging optimization landscapes due to their high density and large macro counts (1,024 selected modules). As presented in Table 7, OrderPlace demonstrates robust scalability compared to existing SOTA methods.

*Table 7.* HPWL ($\times 10^5$) obtained from 5 macro placement methods on "bigblue2" and "bigblue4" circuits. We select 1,024 modules (Deng et al., 2025) as macros for "bigblue2" and "bigblue4". The best results are marked in **bold**.

| Benchmark | MaskPlace | Chipformer | WireMask-EA | EfficientPlace | EGPlace | OrderPlace |
|-----------|-----------|------------|-------------|----------------|---------|------------|
| bigblue2 | $18.64 \pm 0.63$ | $14.06 \pm 0.47$ | $11.63 \pm 0.24$ | $12.20 \pm 0.29$ | $10.67 \pm 0.66$ | $\mathbf{7.78 \pm 0.27}$ |
| bigblue4 | $117.96 \pm 5.62$ | $120.66 \pm 8.03$ | $84.71 \pm 3.94$ | $86.86 \pm 3.41$ | $63.90 \pm 2.30$ | $\mathbf{62.56 \pm 1.87}$ |

## F. Complete Macro Placement Order Strategy Formulations

This appendix provides the complete mathematical formulations for all placement sequence strategies discovered through our LLM-guided evolutionary search framework. We organize strategies by dataset and include both LLM-generated and built-in strategies for completeness. Table 8 summarizes the top-4 strategies for each dataset.

### F.1. Notation and Preliminaries

We first establish the notation used throughout this appendix:

- $I = (M, E, \mathcal{G})$: Macro placement Instance.

*Table 8.* Complete Summary of All Top-4 Strategies

| Dataset | Rank | Strategy | Source | Key Mechanism |
|---|---|---|---|---|
| adaptec1 | 1 | gravity_entropy | LLM | Gravity-entropy hybrid |
| | 2 | field_potential | Built-in | Field attraction model |
| | 3 | hamiltonian | Built-in | Energy minimization |
| | 4 | degree_area_desc | Built-in | Lexicographic sorting |
| adaptec2 | 1 | adaptive_cluster_entropy | LLM | 3-phase adaptive (cluster→entropy) |
| | 2 | adaptive_entropy | LLM | Phase-adaptive entropy with criticality |
| | 3 | entropy | Built-in | Information gain maximization |
| | 4 | critical_path_entropy | LLM | Cascade effect + momentum |
| adaptec3 | 1 | resonance_clustering | LLM | Harmonic resonance + wavefront |
| | 2 | quantum_clustering | LLM | Quantum affinity with bell-curve partial |
| | 3 | adaptive_clustering | LLM | Immediate vs. future balance |
| | 4 | magnetic_criticality | LLM | Magnetic field + net criticality |
| adaptec4 | 1 | spatial_entropy_gravity | LLM | 4-phase spatial-aware + boost |
| | 2 | field_potential | Built-in | Field attraction model |
| | 3 | entropy | Built-in | Information gain maximization |
| | 4 | adaptive_spatial_entropy | LLM | Spatial clustering + cluster bonus |
| bigblue1 | 1 | gravitational_cluster | LLM | Temperature decay + constraint urgency |
| | 2 | adaptive_gravity | LLM | Cluster density variance-based |
| | 3 | spring_potential | Built-in | Spring potential model |
| | 4 | hamiltonian | Built-in | Energy minimization |
| bigblue2 | 1 | adaptive_gravity | LLM | Net-quality weighted $(1/|e|)$ |
| | 2 | field_potential | Built-in | Field attraction model |
| | 3 | gravitational_clustering | LLM | HPWL collapse potential |
| | 4 | adaptive_clustering | LLM | Spatial compactness |
| bigblue3 | 1 | adaptive_clustering | LLM | Net criticality + spatial tightness |
| | 2 | gravity_well | LLM | 3-phase degree weight (3→8→20) |
| | 3 | field_potential | Built-in | Field attraction model |
| | 4 | connectivity_aware_dynamic | LLM | Normalized by $|P_t|$ |
| bigblue4 | 1 | net_closure | LLM | Aggressive closure $(e^{-|e|/10}, r_e^2)$ |
| | 2 | spatial_entropy | LLM | Distance normalization |
| | 3 | chain_formation | LLM | 3-phase chain building |
| | 4 | adaptive_entropy_lookahead | LLM | 4-phase + cascade lookahead |

- $M = \{m_1, ..., m_l\}$: Set of all macros to be placed.

- $E = \{e_1, ..., e_k\}$: Set of $k$ nets.

- $P_t \subseteq M$: Set of macros already placed at step $t$

- $\mathcal{B}_t$: Each net bounding box at step $t$.

- $R_t = M \setminus P_t$: Set of remaining (unplaced) macros

- $\mathcal{N}(m_i)$: Set of nets connected to macro $m_i$

- $N(m_i, m_j)$: Set of nets shared between macros $m_i$ and $m_j$

- $d_i$: Degree (number of connected nets) of macro $m_i$

- $Ar_i$: Area of macro $m_i$

- $(w_i, h_i)$: Width and height of macro $m_i$

- $(x_i, y_i)$: Location of $m_i$

- $\Pi = (\pi_1, \pi_2, \ldots, \pi_l)$: Macro placing sequence.

- $\rho = |P_t|/|M|$: Placement progress ratio

- $|e|$: Size (number of nodes) of net $e$

- $r_e = |e \cap P_t|/|e|$: Completion ratio of net $e$

- $u_e = |e| - |e \cap P_t| - 1$: Number of unplaced nodes in net $e$ excluding current candidate

- $c_i^{\text{det}} = \sum_{e \in \mathcal{N}(m_i)} |e \cap P_t|$: Determined connections

- $c_i^{\text{undet}} = \sum_{e \in \mathcal{N}(m_i)} (|e \cap R_t| - 1)$: Undetermined connections (excluding $m_i$)

- $g$: Grid size; $g_n$: Number of grids

- $\text{Recent}_K(P_t)$: Last $K$ placed macros

The priority function $\phi(m_i, \mathcal{C}_t)$ determines placement order, where lower values indicate higher priority (placed earlier). The next macro to place is:

$$m^* = argmin_{m_i \in R_t} \phi(m_i, \mathcal{C}_t) \tag{20}$$

## F.2. Adaptec1 Strategies

### gravity_entropy (Rank 1, LLM-Generated)

*Description:* Hybrid gravity-entropy model combining gravitational attraction from placed macros with information gain metrics.

*Initial Placement ($P_t = \emptyset$):*

$$\phi(m_i) = \sqrt{d_i \cdot Ar_i} \cdot 1000 \tag{21}$$

*Dynamic Placement:*

$$\phi(m_i, \mathcal{C}_t) = - \left( 0.3 \cdot S_{grvity_i} + 0.4 \cdot S_{cnet i} + 0.2 \cdot S_{pnet_i} + 0.1 \cdot \sqrt{d_i \cdot Ar_i} \right) \tag{22}$$

*Component Definitions:*

1. *Gravity Score* — Attraction from placed macros:

$$S_{grvity_i} = \sum_{m_j \in P_t} |N(m_i, m_j)|^{1.5} \tag{23}$$

2. *Net Closure Score* — Reward for completing/nearly completing nets:

$$S_{cnet i} = \sum_{e \in \mathcal{N}(m_i): u_e = 0} 10 \ln(|e| + 1) + \sum_{e \in \mathcal{N}(m_i): u_e \leq 2} 5 \ln(|e| + 1) \tag{24}$$

3. *Partial Net Score* — Progress on partially placed nets:

$$S_{pnet_i} = \sum_{e \in \mathcal{N}(m_i): |e \cap P_t| > 0} r_e (1 - r_e) \cdot 4 \tag{25}$$

### field_potential (Rank 2, Built-in)

*Description:* Field potential model where placed macros generate attractive field.

Same as field_potential in Table 1.

### hamiltonian (Rank 3, Built-in)

*Description:* Energy minimization model inspired by Hamiltonian mechanics.

Same as hamiltonian in Table 1.

### degree_area_desc (Rank 4, Built-in)

*Description:* Static lexicographic ordering by degree then area.

Same as degree_area_desc in Table 1.

### F.3. Adaptec2 Strategies

### adaptive_cluster_entropy (Rank 1, LLM-Generated)

*Description:* Three-phase adaptive clustering with entropy reduction, prioritizing critical net reduction and local clustering strength.

*Initial Placement:*

$$\phi(m_i) = -(d_i \cdot 100 + Ar_i \cdot 0.001) \tag{26}$$

*Dynamic Placement:*

$$\phi(m_i, \mathcal{C}_t) = -\big(\alpha(\rho) \cdot S_{\text{critical}\,i} + \beta(\rho) \cdot S_{\text{cluster}\,i} + \sigma(\rho) \cdot S_{\text{entropy}\,i} + \gamma(\rho) \cdot d_i\big) \tag{27}$$

*Phase-Adaptive Weights:*

$$(\alpha, \beta, \sigma, \gamma) = \begin{cases} (3.0, 5.0, 2.0, 1.0) & \text{if } \rho < 0.3 \quad \text{(Early)} \\ (4.0, 3.0, 4.0, 0.5) & \text{if } 0.3 \le \rho < 0.7 \quad \text{(Middle)} \\ (5.0, 2.0, 6.0, 0.2) & \text{if } \rho \ge 0.7 \quad \text{(Late)} \end{cases} \tag{28}$$

*Component Definitions:*

1. *Critical Net Score* — Weighted by net size (smaller nets more critical):

$$S_{\text{critical}\,i} = \sum_{e \in \mathcal{N}(m_i):|e \cap P_t|>0} \frac{|e \cap P_t|}{|e| - 1} \cdot \frac{10}{|e| - 1} \tag{29}$$

2. *Cluster Strength* — Quadratic scaling of shared connections:

$$S_{\text{cluster}\,i} = \sum_{m_j \in P_t:|N(m_i, m_j)|>0} |N(m_i, m_j)|^2 \tag{30}$$

3. *Entropy Reduction Score*:

$$S_{\text{entropy}\,i} = \frac{c_i^{\text{det}}}{\sqrt{c_i^{\text{undet}} + 1}} \cdot d_i \tag{31}$$

### adaptive_entropy (Rank 2, LLM-Generated)

*Description:* Enhanced entropy with net criticality and adaptive phase weighting.

*Initial Placement:*

$$\phi(m_i) = -(d_i \cdot 100 + Ar_i \cdot 0.001) \tag{32}$$

*Dynamic Placement:*

$$\phi(m_i, \mathcal{C}_t) = -\left(\alpha(\rho) \cdot S_{\text{conn}\,i} + \beta(\rho) \cdot S_{\text{critical}\,i} - \gamma(\rho) \cdot S_{\text{diff}\,i}\right) \tag{33}$$

*Phase-Adaptive Weights:*

$$(\alpha, \beta, \gamma) = \begin{cases} (1.0, 0.3, 0.5) & \text{if } \rho < 0.3 \\ (0.8, 0.8, 0.3) & \text{if } 0.3 \leq \rho < 0.7 \\ (0.6, 1.5, 0.1) & \text{if } \rho \geq 0.7 \end{cases} \tag{34}$$

*Component Definitions:*

$$S_{\text{conn}\,i} = \frac{c_i^{\text{det}}}{c_i^{\text{undet}} + 1} \cdot d_i \tag{35}$$

$$S_{\text{critical}\,i} = \sum_{e \in \mathcal{N}(m_i):0<u_e\leq2} (3 - u_e) \cdot 20 \tag{36}$$

$$S_{\text{diff}\,i} = Ar_i \cdot 0.0001 \cdot (1 + 2\rho) \tag{37}$$

### entropy (Rank 3, Built-in)

*Description:* Information gain maximization through entropy reduction.

Same as entropy in Table 1.

### critical_path_entropy (Rank 4, LLM-Generated)

*Description:* Critical path entropy prioritizing macros that maximize constraint resolution momentum with cascading information gain.

*Initial Placement:*
$$\phi(m_i) = - \left( d_i^2 \cdot 10 + Ar_i \cdot 0.001 - |\ln(w_i/h_i)| \cdot 50 \right) \tag{38}$$

*Dynamic Placement (Six Components):*

1. *Determined Score*:

$$S_{\text{det}\,i} = \sum_{e \in \mathcal{N}(m_i):|e\cap P_t|>0} \frac{(|e \cap P_t|)^2}{\sqrt{|e|}} \cdot 10 \tag{39}$$

2. *Critical Net Score*:

$$S_{\text{crit}\,i} = \sum_{e \in \mathcal{N}(m_i):0<|e\cap P_t|<|e|-1} r_e^2 \cdot 20 \tag{40}$$

3. *Cascade Potential*:

$$S_{\text{cascade}\,i} = \sum_{m_j \in R_t \setminus \{m_i\}:|N(m_i,m_j)|>0} |N(m_i, m_j)| \cdot \sqrt{d_j} \cdot 0.5 \tag{41}$$

4. *Momentum* (nets nearly complete):

$$S_{\text{mom}\,i} = \sum_{e \in \mathcal{N}(m_i):0<u_e\leq2} (3 - u_e) \cdot 15 \tag{42}$$

5. *Connectivity Strength*:

$$S_{\text{conn}\,i} = \sum_{m_j \in P_t:|N(m_i,m_j)|>0} \mathbf{e}^{0.3|N(m_i,m_j)|} \cdot 2 \tag{43}$$

6. *Degree Urgency*:

$$S_{\text{deg}\,i} = d_i \cdot \ln(d_i + 2) \cdot (1 + 0.5\rho) \tag{44}$$

*Freedom Penalty:*

$$F = 0.5 \ln(|\text{Valid}(P_t, m_i)| + 1) \tag{45}$$

*Phase-Adaptive Combination:*

$$\phi(m_i, \mathcal{C}_t) = \begin{cases} -(3.5S_{\text{det}\,i} + 2.0S_{\text{cascade}\,i} + 1.5S_{\text{conn}\,i} + 1.0S_{\text{deg}\,i} - 0.3c_i^{\text{undet}}) & \rho < 0.25 \\ -(2.5S_{\text{det}\,i} + 4.0S_{\text{crit}\,i} + 3.0S_{\text{mom}\,i} + 2.0S_{\text{conn}\,i} + 1.0S_{\text{cascade}\,i} - 0.5c_i^{\text{undet}} - 0.3F) & 0.25 \leq \rho < 0.6 \\ -(4.5S_{\text{crit}\,i} + 4.0S_{\text{mom}\,i} + 3.0S_{\text{conn}\,i} + 1.5S_{\text{det}\,i} - F + 0.5S_{\text{deg}\,i}) & \rho \geq 0.6 \end{cases} \tag{46}$$

## F.4. Adaptec3 Strategies

### resonance_clustering (Rank 1, LLM-Generated)

*Description:* Harmonic resonance model where macros resonate with placed clusters through harmonic connectivity patterns, with adaptive frequency tuning and net tension minimization.

*Initial Placement:*

$$\phi(m_i) = -(d_i \cdot 120 + Ar_i \cdot 0.0008) \tag{47}$$

*Dynamic Placement (Seven Components):*

$$\phi(m_i, \mathcal{C}_t) = -(1.2S_{R_i} + 1.8S_{C_i} + 1.3S_{T_i} + 1.1S_{W_i} + S_{D_i} + Ar_i^*) + S_{I_i} \tag{48}$$

*Component Definitions:*

1. *Resonance Amplitude*:

$$S_{R_i} = \sum_{m_j \in P_t : |N(m_i, m_j)| > 0} |N(m_i, m_j)|^{1.4} \cdot (1 + 4\rho^{0.8}) \cdot 10 \tag{49}$$

2. *Harmonic Clustering Bonus*:

$$C_{\text{cluster}} = n_{\text{conn}}^{1.3} \cdot 18 \cdot (1 + \rho), \quad n_{\text{conn}} = |\{m_j \in P_t : |N(m_i, m_j)| > 0\}| \tag{50}$$

3. *Critical Net Closure*:

$$S_{C_i} = \sum_{e \in \mathcal{N}(m_i) : u_e = 0} 70 \cdot (1 + 1.5\rho) \tag{51}$$

4. *Net Tension* (high completion nets):

$$S_{T_i} = \sum_{e \in \mathcal{N}(m_i) : r_e > 0.7} r_e^3 \cdot 35 \cdot |e| + \sum_{e \in \mathcal{N}(m_i) : r_e \geq 0.4} r_e^2 \cdot 20 \cdot |e| \tag{52}$$

5. *Wavefront Coherence* (last 8 placed macros):

$$S_{W_i} = \sum_{m_j \in \text{Recent}_8(P_t)} |N(m_i, m_j)|^{1.2} \cdot 12 + \mathbf{1}[n_{\text{wave}} \geq 3] \cdot n_{\text{wave}} \cdot 25 \tag{53}$$

where $n_{\text{wave}} = |\{m_j \in \text{Recent}_8(P_t) : |N(m_i, m_j)| > 0\}|$.

6. *Degree Factor* (exponential decay):

$$S_{D_i} = d_i \cdot 6 \cdot \exp(-2\rho) \tag{54}$$

7. *Area Factor* (three-phase):

$$S_{Ar_i^*} = \begin{cases} \sqrt{Ar_i} \cdot 0.004 & \rho < 0.25 \\ \sqrt{Ar_i} \cdot 0.001 & 0.25 \leq \rho < 0.75 \\ -\sqrt{Ar_i} \cdot 0.003 & \rho \geq 0.75 \end{cases} \tag{55}$$

8. *Isolation Penalty*:

$$S_{I_i} = \mathbf{1}[n_{\text{conn}} = 0 \wedge \rho > 0.4] \cdot 100 \cdot \rho^2 \tag{56}$$

## quantum_clustering (Rank 2, LLM-Generated)

*Description:* Quantum-inspired clustering with probabilistic affinity fields that strengthen with placement, prioritizing net closure and spatial coherence.

*Initial Placement:*

$$\phi(m_i, \mathcal{C}_t) = -\left(S_{\text{affinity}_i} + 1.5 S_{\text{closure}_i} + S_{\text{partial}_i} + S_{\text{recent}_i} + S_{D_i} + S_{Ar_i^*} + S_{\text{density}_i}\right) \tag{57}$$

*Component Definitions:*

1. *Quantum Affinity*:

$$S_{\text{affinity}_i} = \sum_{m_j \in P_t : |N(m_i, m_j)| > 0} |N(m_i, m_j)|^{1.3} \cdot 12 \cdot (1 + 3\rho^{0.7}) \tag{58}$$

2. *Net Closure Bonus*:

$$S_{\text{closure}_i} = \sum_{e \in \mathcal{N}(m_i) : u_e = 0} 50 \cdot (1 + \rho) \tag{59}$$

3. *Partial Net Bonus* (Bell curve at 65% completion):

$$S_{\text{partial}_i} = \sum_{e \in \mathcal{N}(m_i) : |e \cap P_t| > 0} \exp\left(-\frac{(r_e - 0.65)^2}{0.08}\right) \cdot 25 \cdot |e| \tag{60}$$

4. *Recent Connection Bonus* (last 5 placed):

$$S_{\text{recent}_i} = \sum_{m_j \in \text{Recent}_5(P_t)} |N(m_i, m_j)| \cdot 8 \tag{61}$$

5. *Degree Factor*:

$$S_{D_i} = d_i \cdot 5 \cdot (1 - 0.7\rho) \tag{62}$$

6. *Area Factor*:

$$S_{Ar_i^*} = \begin{cases} \sqrt{Ar_i} \cdot 0.003 & \rho < 0.3 \\ 0 & 0.3 \leq \rho < 0.7 \\ -\sqrt{Ar_i} \cdot 0.002 & \rho \geq 0.7 \end{cases} \tag{63}$$

7. *Density Bonus/Penalty*:

$$S_{\text{density}_i} = \begin{cases} n_{\text{conn}} \cdot 15 \cdot (1 + \rho) & n_{\text{conn}} > 0 \\ -50 \cdot \rho^2 & n_{\text{conn}} = 0 \end{cases} \tag{64}$$

where $n_{\text{conn}} = |\{m_j \in P_t : |N(m_i, m_j)| > 0\}|$.

## adaptive_clustering (Rank 3, LLM-Generated)

*Description:* Multi-scale connectivity balancing immediate placement benefit with future cluster cohesion.

*Initial Placement:*

$$\phi(m_i) = -(d_i \cdot 100 + Ar_i \cdot 0.001) \tag{65}$$

*Dynamic Placement:*

$$\phi(m_i, \mathcal{C}_t) = -\left(\alpha(\rho) \cdot S_{\text{imm}_i} + \beta(\rho) \cdot S_{\text{future}_i} + \gamma(\rho) \cdot d_i + S_{I_{\text{penalty}_i}}\right) \tag{66}$$

*Phase-Adaptive Weights:*

$$(\alpha, \beta, \gamma) = \begin{cases} (0.6, 0.3, 0.1) & \rho < 0.4 \\ (0.8, 0.1, 0.1) & 0.4 \leq \rho < 0.7 \\ (0.9, 0.0, 0.1) & \rho \geq 0.7 \end{cases} \tag{67}$$

*Component Definitions:*

$$S_{\text{imm}\,i} = \sum_{m_j \in P_t} |N(m_i, m_j)| \cdot 15 \tag{68}$$

$$S_{\text{future}\,i} = \sum_{e \in \mathcal{N}(m_i)} \sum_{m_j \in e \cap R_t, m_j \neq m_i} 5 \tag{69}$$

$$S_{I_{\text{penalty}\,i}} = \mathbf{1}\left[ \frac{S_{\text{imm}\,i}}{15} + n_{\text{unplaced}} < 2 \right] \cdot (-50) \tag{70}$$

where $n_{\text{umplaced}} = \sum_{e \in \mathcal{N}(m_i)} |e \cap R_t \setminus \{m_i\}|$.

### magnetic_criticality (Rank 4, LLM-Generated)

*Description:* Magnetic field model where placed macros create magnetic field, prioritizing macros that resolve critical nets.

*Initial Placement:*

$$\phi(m_i) = -d_i \cdot 1000 \tag{71}$$

*Dynamic Placement:*

$$\phi(m_i, \mathcal{C}_t) = S_{F_m\,i} + S_{\text{crit}\,i} + S_{d\,i} + S_{A\,i} \tag{72}$$

*Component Definitions:*

$$S_{F_m\,i} = -\sum_{m_j \in P_t : |N(m_i, m_j)| > 0} |N(m_i, m_j)|^2 \cdot 15 \tag{73}$$

$$S_{\text{crit}\,i} = -\sum_{e \in \mathcal{N}(m_i) : 0 < u_e \leq 3} r_e^2 \cdot 50 \tag{74}$$

$$S_{d\,i} = -d_i \cdot 3 \cdot (1 - 0.7\rho) \tag{75}$$

$$S_{A\,i} = \begin{cases} -Ar_i \cdot 5 \times 10^{-5} & \rho < 0.5 \\ Ar_i \cdot 3 \times 10^{-5} & \rho \geq 0.5 \end{cases} \tag{76}$$

## F.5. Adaptec4 Strategies

### spatial_entropy_gravity (Rank 1, LLM-Generated)

*Description:* Entropy-driven with spatial gravity pull and net tension awareness, featuring four-phase adaptive prioritization.

*Initial Placement:*

$$\phi(m_i) = -(d_i \cdot \sqrt{Ar_i}) \tag{77}$$

*Dynamic Placement:*

$$\phi(m_i, \mathcal{C}_t) = -\left( \alpha_1 S_{e_i} + \alpha_2 S_{G_i} + \alpha_3 S_{T_i} + \alpha_4 \sqrt{d_i} + \alpha_5 \ln(Ar_i + 1) \right) \cdot \text{boost} \tag{78}$$

*Four-Phase Weights:*

$$(\alpha_1, \alpha_2, \alpha_3, \alpha_4, \alpha_5) = \begin{cases} (1.0, 0.3, 0.5, 2.0, 0.5) & \rho < 0.25 \\ (2.0, 1.5, 1.0, 1.0, 0.3) & 0.25 \leq \rho < 0.5 \\ (1.5, 3.0, 2.0, 0.5, 0.1) & 0.5 \leq \rho < 0.75 \\ (1.0, 4.0, 3.0, 0.3, 0.05) & \rho \geq 0.75 \end{cases} \tag{79}$$

*Component Definitions:*

1. *Entropy Score*:

$$S_{e_i} = \sum_{e \in \mathcal{N}(m_i):|e \cap P_t|>0} \begin{cases} \frac{|e \cap P_t|}{\sqrt{u_e+1}} & u_e > 0 \\ |e \cap P_t| \cdot 2 & u_e = 0 \text{ (net complete)} \end{cases} \tag{80}$$

2. *Spatial Gravity* (with net criticality):

$$S_{G_i} = \sum_{e \in \mathcal{N}(m_i):|e \cap P_t|>0} |e \cap P_t| \cdot \kappa(e) \tag{81}$$

3. *Net Tension*:

$$S_{T_i} = \sum_{e \in \mathcal{N}(m_i)} \begin{cases} 20 \cdot \kappa(e) & r_e \geq 0.8 \\ 10 \cdot \kappa(e) & r_e \geq 0.6 \\ 5 \cdot \kappa(e) & r_e \geq 0.4 \\ 2 \cdot \kappa(e) & |e \cap P_t| \geq 2 \end{cases} \tag{82}$$

where $\kappa(e) = \frac{1}{\sqrt{|e|}}$.

*Boost Factor:*

$$\text{boost} = \begin{cases} 1.56 & S_{e_i} > 10 \\ 1.2 & S_{e_i} > 5 \\ 1.0 & \text{otherwise} \end{cases} \tag{83}$$

**field_potential (Rank 2, Built-in)**

Same as field_potential in Table 1.

**entropy (Rank 3, Built-in)**

Same as entropy in Table 1.

**adaptive_spatial_entropy (Rank 4, LLM-Generated)**

*Description:* Enhanced entropy with spatial clustering and net criticality awareness.

*Initial Placement:*

$$\phi(m_i) = -(d_i \cdot \sqrt{Ar_i}) \tag{84}$$

*Dynamic Placement (Three-Phase):*

$$\phi(m_i, \mathcal{C}_t) = \begin{cases} -S_{e_i} \cdot d_i \cdot (1 + 0.1\sqrt{Ar_i}) & \rho < 0.3 \\ -S_{e_i} \cdot (d_i + 10S_{\text{spatial}_i}) \cdot (1 + 0.5n_{\text{crit}}) & 0.3 \leq \rho < 0.7 \\ -S_{e_i} \cdot (20S_{\text{spatial}_i} + 0.5d_i) \cdot (1 + n_{\text{crit}}) & \rho \geq 0.7 \end{cases} \tag{85}$$

where $n_{\text{crit}} = |\{e \in \mathcal{N}(m_i) : |e \cap P_t| \geq 2 \wedge |e| \geq 3\}|$.

*Component Definitions:*

$$S_{e_i} = \begin{cases} \frac{c_i^{\text{det}}}{\ln(c_i^{\text{undet}}+|e|)} & c_i^{\text{undet}} > 0 \\ 2 \cdot c_i^{\text{det}} & c_i^{\text{undet}} = 0 \end{cases} \tag{86}$$

$$S_{\text{spatial}_i} = \sum_{e \in \mathcal{N}(m_i):|e \cap P_t|>0} \frac{|e \cap P_t|}{|e \cap P_t| + |e \cap R_t| + 1} \tag{87}$$

*Cluster Bonus:*

$$\phi_{\text{final}}(m_i, \mathcal{C}_t) = \phi(m_i, \mathcal{C}_t) \cdot (1 + 0.1 \cdot c_i^{\text{det}}) \quad \text{if } c_i^{\text{det}} \geq 3 \tag{88}$$

## F.6. BigBlue1 Strategies

### gravitational_cluster (Rank 1, LLM-Generated)

*Description:* Gravitational model with adaptive clustering where strong nets create gravity wells, prioritizing constrained macros.

*Initial Placement:*

$$\phi(m_i) = -(d_i \cdot \sqrt{Ar_i}) \tag{89}$$

*Dynamic Placement:*

*Temperature (simulated annealing):*

$$S_{T_i} = \exp(-3\rho) \tag{90}$$

*Gravitational Force:*

$$S_{F_i} = 100 \cdot \sum_{m_j \in P_t : |N(m_i, m_j)| > 0} |N(m_i, m_j)|^{1.5} \tag{91}$$

*Cluster Benefit:*

$$S_{\text{cluster}i} = \begin{cases} n_{\text{placed}} \cdot 50 - n_{\text{unplaced}} \cdot 10 & 0.3 \le \rho < 0.7 \\ n_{\text{placed}} \cdot 30 & \rho \ge 0.7 \end{cases} \tag{92}$$

where $n_{\text{placed}} = \sum_{e \in \mathcal{N}(m_i)} |e \cap P_t \setminus \{m_i\}|$, $n_{\text{unplaced}} = \sum_{e \in \mathcal{N}(m_i)} |e \cap R_t \setminus \{m_i\}|$.

*Constraint Urgency:*

$$S_{U_i} = \frac{1000}{f(P_t, m_i)}, \quad f = \max\left(1, (g_n - \lceil w_i/g \rceil)(g_n - \lceil h_i/g \rceil) - 0.5|P_t|\right) \tag{93}$$

*Phase-Adaptive Combination:*

$$\phi(m_i, \mathcal{C}_t) = \begin{cases} -S_{F_i} & \rho < 0.3 \\ -(S_{F_i} + S_{\text{cluster}i}) & 0.3 \le \rho < 0.7 \\ -(S_{F_i} + S_{\text{cluster}i} + S_{U_i} \cdot S_{T_i}) & \rho \ge 0.7 \end{cases} \tag{94}$$

### adaptive_gravity (Rank 2, LLM-Generated)

*Description:* Adaptive gravity model where placed macros exert gravitational pull weighted by connectivity and cluster density.

*Initial Placement:*

$$\phi(m_i) = -(d_i \cdot 100 + Ar_i \cdot 0.001) \tag{95}$$

*Dynamic Placement (No Connections):*

$$\phi(m_i) = -(d_i \cdot 10 + Ar_i \cdot 0.0001) \quad \text{if no connected placed macros} \tag{96}$$

*Dynamic Placement (With Connections):*

$$\phi(m_i, \mathcal{C}_t) = S_{F_i} + S_{d_i} + S_{\text{cluster}i} + S_{U_i} \tag{97}$$

*Gravitational Force:*

$$S_{F_i} = -\sum_{(m_i, m_j) \in e} \frac{|N(m_i, m_j)| \cdot d_j}{(1/(|N(m_i, m_j)| + 0.1))^2} \tag{98}$$

*Other Components:*

$$S_{d_i} = -d_i \cdot 5 \quad \text{(Degree Bonus)} \tag{99}$$

$$S_{\text{cluster}_i} = -\frac{100}{\text{Var}(weight) + 1} \quad \text{(Cluster Density)} \tag{100}$$

$$S_{U_i} = -n_{\text{partial}} \cdot \rho \cdot 50 \quad \text{(Urgency Factor)} \tag{101}$$

where $weight = [-\sum_{(m_1, m_j) \in e} |N(m_1, m_j)|, -\sum_{(m_2, m_j) \in e} |N(m_2, m_j)|, \ldots]$ are connection weights to placed macros, and $n_{\text{partial}} = |\{e \in \mathcal{N}(m_i) : 0 < |e \cap P_t| < |e| - 1\}|$.

### spring_potential (Rank 3, Built-in)

*Description:* Spring potential model where connections act as springs.

Same as spring_potential in Table 1.

### hamiltonian (Rank 4, Built-in)

Same as hamiltonian in Table 1.

### F.7. BigBlue2 Strategies

### adaptive_gravity (Rank 1, LLM-Generated)

*Description:* Gravitational model where macros attract each other through shared nets, with adaptive strength based on placement density.

*Initial Placement:*

$$\phi(m_i) = -(d_i \cdot \sqrt{Ar_i}) \tag{102}$$

*Dynamic Placement (No Connections):*

$$\phi(m_i) = 1 \times 10^9 \quad \text{(very low priority)} \tag{103}$$

*Dynamic Placement:*

$$\phi(m_i, \mathcal{C}_t) = -(S_{F_i} + S_{\text{complete}_i}) \tag{104}$$

*Gravitational Force:*

$$S_{F_i} = \sum_{m_j \in P_t : |N(m_i, m_j)| > 0} |N(m_i, m_j)| \cdot \sqrt{Ar_i \cdot A_j} \cdot Q_{\text{net}} \cdot (1 + 2\rho) \tag{105}$$

*Net Quality* (smaller nets are higher quality):

$$Q_{\text{net}} = \sum_{e \in N(m_i, m_j)} \frac{1}{|e|} \tag{106}$$

*Completion Bonus:*

$$S_{\text{complete}_i} = \sum_{e \in \mathcal{N}(m_i) : |e \cap P_t| > 1, |e| > 2} r_e \cdot |e \cap P_t| \cdot 50 \tag{107}$$

### field_potential (Rank 2, Built-in)

Same as field_potential in Table 1.

### gravitational_clustering (Rank 3, LLM-Generated)

*Description:* Gravitational model with HPWL awareness.

*Initial Placement ($|P_t| < 3$):*

$$\phi(m_i) = -(d_i \cdot Ar_i) \tag{108}$$

*Dynamic Placement:*

$$\phi(m_i, \mathcal{C}_t) = -\big(15S_{F_i} + 8S_{\text{collapse}_i} + 2d_i\big) \tag{109}$$

*Component Definitions:*

$$S_{F_i} = \sum_{m_j \in P_t : |N(m_i, m_j)| > 0} |N(m_i, m_j)|^{1.5} \tag{110}$$

$$S_{\text{collapse}_i} = \sum_{e \in \mathcal{N}(m_i) : e \in \mathcal{B}, |e \cap P_t| > 0} |e \cap P_t| \cdot 2 \tag{111}$$

## adaptive_clustering (Rank 4, LLM-Generated)

*Description:* Adaptive clustering prioritizing macros that form tight spatial clusters.

*Initial Placement:*

$$\phi(m_i) = -d_i \cdot 1000 \tag{112}$$

*Dynamic Placement:*

$$\phi(m_i, \mathcal{C}_t) = -(S_{\text{cluster}i} + S_{ei} + d_i \cdot 10) \tag{113}$$

*Cluster Score:*

$$S_{\text{cluster}i} = n_{\text{shared}i} \cdot \text{compactness} \cdot 100 \tag{114}$$

where $n_{\text{shared}i}$ indicates how many nets simultaneously contain the macro $m_i$, and compactness $= \frac{1}{1 + \sqrt{\text{Var}(x) + \text{Var}(y)}/g}$.

*Entropy Score:*

$$S_{ei} = \begin{cases} \frac{c_i^{\text{det}}}{c_i^{\text{undet}} + 1} \cdot 50 & c_i^{\text{undet}} > 0 \\ c_i^{\text{det}} \cdot 50 & c_i^{\text{undet}} = 0 \end{cases} \tag{115}$$

## F.8. BigBlue3 Strategies

## adaptive_clustering (Rank 1, LLM-Generated)

*Description:* Dynamic strategy prioritizing macros that form tight clusters with placed macros, weighted by net criticality.

*Initial Placement:*

$$\phi(m_i) = -(d_i \cdot Ar_i + d_i \cdot 1000) \tag{116}$$

*Dynamic Placement:*

$$\phi(m_i, \mathcal{C}_t) = -S_{\text{conn}i} - d_i \cdot S_d(\rho) \tag{117}$$

*Connectivity Score:*

$$S_{\text{conn}i} = \sum_{e \in \mathcal{N}(m_i) : |e \cap P_t| > 0} \frac{|e \cap P_t|}{\ln(|e| + 2)} \cdot (1 + 2r_e) \cdot 100 \tag{118}$$

*Spatial Tightness Boost:*

$$S_{\text{conn}i} \mathrel{*}= (1 + \text{cluster\_tightness}), \quad \text{cluster\_tightness} = \frac{1}{1 + \sqrt{\text{Var}(x) + \text{Var}(y)}/1000} \tag{119}$$

*Degree Weight:*

$$S_d(\rho) = \begin{cases} 2 & \rho < 0.5 \\ 5 & \rho \geq 0.5 \end{cases} \tag{120}$$

### gravity_well (Rank 2, LLM-Generated)

*Description:* Gravity well model guiding compact placement.

*Initial Placement:*

$$\phi(m_i) = -(d_i \cdot Ar_i + d_i^2 \cdot 10) \tag{121}$$

*Dynamic Placement:*

$$\phi(m_i, \mathcal{C}_t) = -\left(S_{F_i} \cdot 100 + d_i \cdot S_d(\rho) + 0.5\ln(Ar_i + 1)\right) \tag{122}$$

*Net Weight:*

$$S_{\text{net}}(e) = \frac{1}{\sqrt{|e| + 1}} \cdot (1 + 3r_e) \tag{123}$$

*Gravity with Compactness:*

$$S_{F_i} = \left(\sum_{m_j \in e} S_{\text{net}}(e)\right) \cdot (1 + 2 \cdot \text{compactness}) \tag{124}$$

*Compactness:*

$$\text{compactness} = \frac{1}{1 + \sqrt{\text{Var}(x) + \text{Var}(y)}/10000} \tag{125}$$

*Degree Weight:*

$$S_d(\rho) = \begin{cases} 3 & \rho < 0.3 \\ 8 & 0.3 \le \rho < 0.7 \\ 20 & \rho \ge 0.7 \end{cases} \tag{126}$$

### field_potential (Rank 3, Built-in)

Same as field_potential in Table 1.

### connectivity_aware_dynamic (Rank 4, LLM-Generated)

*Description:* Normalized connectivity scoring avoiding late-stage bias.

*Initial Placement:*

$$\phi(m_i) = -d_i \cdot 100 - Ar_i \cdot 0.01 \tag{127}$$

*Dynamic Placement:*

$$\phi(m_i, \mathcal{C}_t) = -\frac{S_{\text{conn}i}}{|P_t|} \cdot 1000 - d_i \cdot 10 - Ar_i \cdot 0.01 \tag{128}$$

where:

$$S_{\text{conn}i} = \sum_{e \in \mathcal{N}(m_i)} |e \cap P_t| \tag{129}$$

*Key Feature:* Division by $|P_t|$ normalizes the connectivity score to prevent bias toward later stages.

### F.9. BigBlue4 Strategies

### net_closure (Rank 1, LLM-Generated)

*Description:* Aggressive net closure prioritization for critical nets with high placed/total ratio.

*Initial Placement:*

$$\phi(m_i) = -(d_i \cdot \sqrt{Ar_i}) \tag{130}$$

*Dynamic Placement (No Connections):*

$$\phi(m_i) = 1 \times 10^9 \tag{131}$$

*Dynamic Placement:*

$$\phi(m_i, \mathcal{C}_t) = -\left(2S_{\text{closure}\,i} + S_{\text{conn}\,i} + 3S_{e\,i}\right) \tag{132}$$

*Net Closure Score:*

$$S_{\text{closure}\,i} = \sum_{e \in \mathcal{N}(m_i):|e \cap P_t|>0} \begin{cases} 10 \cdot e^{-|e|/10} & u_e = 0 \text{ (last node)} \\ r_e^2 \cdot e^{-|e|/10} \cdot 3 & \text{otherwise} \end{cases} \tag{133}$$

*Connectivity Strength:*

$$S_{\text{conn}\,i} = \sum_{m_j \in P_t:|N(m_i,m_j)|>0} |N(m_i, m_j)| \cdot \bar{\kappa} \tag{134}$$

where $\bar{\kappa}$ is the average net criticality.

*Entropy Reduction:*

$$S_{e\,i} = \frac{c_i^{\text{det}}}{c_i^{\text{undet}} + 1} \tag{135}$$

## spatial_entropy (Rank 2, LLM-Generated)

*Description:* Enhanced entropy with spatial clustering.

*Initial Placement:*

$$\phi(m_i) = -(d_i \cdot Ar_i) \tag{136}$$

*Dynamic Placement (No Connections):*

$$\phi(m_i) = 1 \times 10^9 \tag{137}$$

*Dynamic Placement:*

$$\phi(m_i, \mathcal{C}_t) = -w \cdot S_{\text{conn}\,i} + 0.3 \cdot dis_{\text{norm}} - 5S_{e\,i} \tag{138}$$

*Weighted Connectivity:*

$$S_{\text{conn}\,i} = \sum_{e \in \mathcal{N}(m_i):|e \cap P_t|>0} |e \cap P_t| \cdot f(e), \quad f(e) = \frac{1}{\sqrt{|e|}} \tag{139}$$

*Normalized Distance:*

$$dis_{\text{norm}} = \frac{\bar{dis}}{\sqrt{2} \cdot g_n \cdot g} \tag{140}$$

where $\bar{dis}$ is the average distance from connected placed macros to grid center.

*Entropy Score:*

$$S_{e\,i} = \frac{|e \cap P_t|}{c_i^{\text{undet}} + 1} \tag{141}$$

## chain_formation (Rank 3, LLM-Generated)

*Description:* Chain formation strategy prioritizing connectivity chains between placed and unplaced macros.

*Initial Placement:*

$$\phi(m_i) = -(d_i \cdot \sqrt{Ar_i}) \tag{142}$$

*Dynamic Placement (No Connections):*

$$\phi(m_i) = 1 \times 10^9 \tag{143}$$

*Dynamic Placement:*

$$\phi(m_i, \mathcal{C}_t) = -(S_{\text{chain}\,i} + S_{\text{crit}\,i} + \ln(d_i + 1)) \tag{144}$$

*Placed/Unplaced Connectivity:*

$$S_{\text{placed}} = \sum_{e \in \mathcal{N}(m_i):|e \cap P_t|>0} |e \cap P_t| \cdot \kappa(e) \tag{145}$$

$$S_{\text{unplaced}} = \sum_{e \in \mathcal{N}(m_i)} |e \cap R_t \setminus \{m_i\}| \cdot \kappa(e) \tag{146}$$

where $\kappa(e) = \exp\{-|e|/5\}$ is net criticality.

*Chain Score (Phase-Adaptive):*

$$S_{\text{chain}\,i} = S_{\text{placed}} \cdot \begin{cases} (1 + 0.8 \cdot S_{\text{unplaced}}) & \rho < 0.3 \\ (1 + 0.5 \cdot S_{\text{unplaced}}) & 0.3 \le \rho < 0.7 \\ (1 + 0.2 \cdot S_{\text{unplaced}}) & \rho \ge 0.7 \end{cases} \tag{147}$$

*Critical Bonus:*

$$S_{\text{crit}\,i} = (S_i^{\text{placed}} + 0.5 \cdot S_i^{\text{unplaced}}) \cdot 2 \tag{148}$$

where $S_i^{\text{placed}} = \sum_{e \in \mathcal{N}(m_i):|e \cap P_t|>0} \kappa(e)$, and $S_i^{\text{unplaced}} = \sum_{e \in \mathcal{N}(m_i):|e \cap R_t \setminus \{m_i\}|>0} \kappa(e)$.

### adaptive_entropy_lookahead (Rank 4, LLM-Generated)

*Description:* Adaptive entropy with lookahead maximizing information gain while predicting cascading placement effects.

*Initial Placement:*

$$\phi(m_i) = -(d_i \cdot \ln(Ar_i + 1)) \tag{149}$$

*Dynamic Placement (No Connections):*

$$\phi(m_i) = 1 \times 10^9 \tag{150}$$

*Dynamic Placement:*

$$\phi(m_i, \mathcal{C}_t) = -(\alpha S_{e\,i} + \beta S_{\text{crit}\,i} + \gamma S_{\text{look}\,i} + \sigma \ln(d_i + 1)) \tag{151}$$

*Four-Phase Weights:*

$$(\alpha, \beta, \gamma, \sigma) = \begin{cases} (3.0, 2.0, 1.5, 1.0) & \rho < 0.25 \\ (2.5, 2.5, 1.2, 1.2) & 0.25 \le \rho < 0.5 \\ (2.0, 3.0, 0.8, 1.5) & 0.5 \le \rho < 0.75 \\ (1.5, 3.5, 0.5, 2.0) & \rho \ge 0.75 \end{cases} \tag{152}$$

*Entropy Reduction:*

$$S_{e\,i} = \begin{cases} \frac{c_i^{\text{det}} \cdot \kappa}{\sqrt{c_i^{\text{undet}}+1}} & c_i^{\text{undet}} > 0 \\ c_i^{\text{det}} \cdot \kappa \cdot 2 & c_i^{\text{undet}} = 0 \end{cases} \tag{153}$$

where $\kappa = \exp\{-|e|/5\}$ is net criticality.

*Critical Bonus:*

$$S_{\text{crit}\,i} = (S_i^{\text{placed}} + 0.5 \cdot S_i^{\text{unplaced}}) \cdot 2 \tag{154}$$

where $S_i^{\text{placed}} = \sum_{e \in \mathcal{N}(m_i):|e \cap P_t|>0} \kappa(e)$, and $S_i^{\text{unplaced}} = \sum_{e \in \mathcal{N}(m_i):|e \cap R_t \setminus \{m_i\}|>0} \kappa(e)$, and $\kappa(e) = \exp\{-|e|/5\}$.

*Lookahead Score (Cascade Effect):*

$$S_{\text{look}\,i} = \sum_{e \in \mathcal{N}(m_i)} \sum_{m_j \in e \cap R_t \setminus \{m_i\}} \exp\{-|e|/5\} \cdot \ln(d_j + 1) \cdot 0.3 \tag{155}$$

## F.10. Dataset-Specific Strategy Insights

Our analysis reveals that different circuit characteristics favor different strategy emphases:

**Adaptec circuits** tend to favor entropy-based strategies with strong net closure components. The best-performing strategies (`gravity_entropy`, `adaptive_cluster_entropy`, `resonance_clustering`, `spatial_entropy_gravity`) all incorporate explicit entropy reduction terms.

**BigBlue circuits** show stronger preference for gravitational/clustering models. The winning strategies (`gravitational_cluster`, `adaptive_gravity`, `adaptive_clustering`, `net_closure`) emphasize spatial clustering and connectivity strength over pure information-theoretic measures.

This suggests that the optimal strategy design depends on circuit topology characteristics such as net size distribution, macro count, and connectivity patterns.

# G. More Results

## G.1. Population Quality Evaluator Analysis.

To validate the fidelity of our Population Quality Evaluator, we conducted a "reductio ad absurdum" experiment on the `adaptec1`, `adaptec3` and `bigblue3` dataset. While our framework typically only executes the full optimization for the top-4 strategies ranked by the evaluator, here we performed full placement optimization for all generated strategies to verify if the evaluator's ranking aligns with the ground truth performance.

**Indicator Selection.** The Population Quality Evaluator generates several statistical metrics for the initial population. We identify the Mean Potential Score (calculated according to formula 4.) as the most representative indicator. As shown in Table 9, the Mean Potential Score exhibits a strong positive correlation with the final converged HPWL. A lower mean score indicates that the strategy consistently produces high-quality initial solutions that are situated in favorable regions of the solution space, thereby facilitating faster and better convergence.

**Correlation Analysis.** The comparison results are presented in Table 9. The evaluator successfully identified `dynamics_field_potential` as the top-ranking strategy (Rank 1), achieving the lowest Mean Potential Score of $7.60 \times 10^5$. Crucially, the ground truth results confirm this prediction: `dynamics_field_potential` ultimately achieved the best final HPWL of $5.80 \times 10^5$.

Furthermore, the evaluator effectively filtered out inferior strategies. For instance, the `random` strategy, which was ranked last by the evaluator (Mean Score $22.4 \times 10^5$), indeed produced the worst final performance. Although there are minor ranking permutations among the middle-tier strategies (e.g., between `degree_area_desc` and `dynamics_spring_potential`), the evaluator accurately distinguishes the "elite tier" from the rest. This validates that selecting the top-ranked strategies based on the Mean Potential Score is a reliable proxy for final placement quality, significantly reducing the computational overhead by avoiding full runs on unpromising candidates.

*Table 9.* Validation of the Population Quality Evaluator on `adaptec1`. The "Proxy Metric" is the Mean Potential Score from the evaluator (lower is better), and "Ground Truth" is the final converged HPWL. The evaluator correctly predicts the best-performing strategy.

| Strategy | Proxy Evaluation (Prediction) | | Full Optimization (Ground Truth) | |
|---|---|---|---|---|
| | Mean Score ($\times 10^5$) | Rank | Final HPWL ($\times 10^5$) | Rank |
| **dynamics_field_potential** | **7.60** | **1** | **5.80** | **1** |
| degree_area_desc | 7.75 | 2 | 5.83 | 2 |
| degree_desc | 7.81 | 3 | 5.87 | 4 |
| net_area_desc | 7.85 | 4 | 5.90 | 5 |
| dynamics_spring_potential | 8.08 | 5 | 5.83 | 3 |
| area_desc | 8.50 | 6 | 6.26 | 6 |
| random | 22.43 | 7 | $> 10.0$ | 7 |

In addition, we have also validated the proxy evaluator on larger-scale benchmarks (`adaptec3` and `bigblue3`). As

shown below tables, the proxy metric remains strongly correlated with final HPWL, correctly identifying the top strategy in both cases.

*Table 10.* Validation of the Population Quality Evaluator on `adaptec3`. The "Proxy Metric" is the Mean Potential Score from the evaluator (lower is better), and "Ground Truth" is the final converged HPWL. The evaluator correctly predicts the best-performing strategy.

| Strategy | Proxy Evaluation (Prediction) | | Full Optimization (Ground Truth) | |
|---|---|---|---|---|
| | Mean Score ($\times 10^5$) | Rank | Final HPWL ($\times 10^5$) | Rank |
| spring_potential | 82.18 | 3 | 53.70 | 1 |
| hamiltonian | 89.79 | 5 | 56.35 | 2 |
| field_potential | 70.68 | 1 | 58.38 | 3 |
| entropy | 73.62 | 2 | 58.99 | 4 |
| net_area_desc | 10.23 | 6 | 59.72 | 5 |
| degree_area_desc | 88.83 | 4 | 62.52 | 6 |
| area_degree_desc | 103.31 | 7 | 63.52 | 7 |

*Table 11.* Validation of the Population Quality Evaluator on `bigblue3`. The "Proxy Metric" is the Mean Potential Score from the evaluator (lower is better), and "Ground Truth" is the final converged HPWL. The evaluator correctly predicts the best-performing strategy.

| Strategy | Proxy Evaluation (Prediction) | | Full Optimization (Ground Truth) | |
|---|---|---|---|---|
| | Mean Score ($\times 10^5$) | Rank | Final HPWL ($\times 10^5$) | Rank |
| field_potential | 102.73 | 1 | 49.72 | 1 |
| gradient_force | 115.28 | 2 | 60.25 | 2 |
| degree_desc | 149.14 | 4 | 66.58 | 3 |
| net_area_desc | 176.30 | 5 | 67.60 | 4 |
| degree_area_desc | 138.24 | 3 | 72.05 | 5 |
| area_degree_desc | 200.51 | 6 | 111.04 | 6 |
| area_desc | 220.05 | 7 | 121.71 | 7 |

## G.2. Parameter Sensitivity Analysis and Ablation Studies.

### G.2.1. THE NUMBER OF PARAMETER TOP-K

For each user prompt, the Top-K elite strategies are provided to the LLM, which helps guide it to generate ranking strategies that better align with the dataset preferences. We analyze the impact of parameter K, and the results are shown in Figure 3. Neither excessively large nor excessively small values of K yield good performance; the best results are achieved when K=3, where the LLM generates the most effective initial strategies.

### G.2.2. EFFECT OF PRIOR KNOWLEDGE IN USER PROMPTS

In addition, we conduct an ablation study in which the manually designed initial macro placement order strategies are removed (the user prompt is omitted). We observe that, without these priors, the number of newly generated effective strategies by the LLM ranges from 1 to 3, whereas it increases to 6–8 when the priors are included. This result demonstrates that such prior knowledge plays a crucial role in optimizing the LLM's preference alignment.

### G.2.3. MULTI-LLM ROBUSTNESS AND CONVERGENCE ANALYSIS

To further investigate the robustness of the proposed strategy generation framework with respect to the choice of LLM backend, we conduct a case study on the Adaptec3 benchmark using three representative LLMs, namely Claude Sonnet 4.5, GPT-4o, and DeepSeek-V3.2. In this experiment, the number of evolutionary generations is extended to 6 in order to analyze the convergence behavior of the search process. The convergence curves of the best proxy HPWL, the elite population update rates, and the composition of the elite strategies are shown in Figure 4. In addition, the final placement quality on different benchmarks is summarized in Table 12.

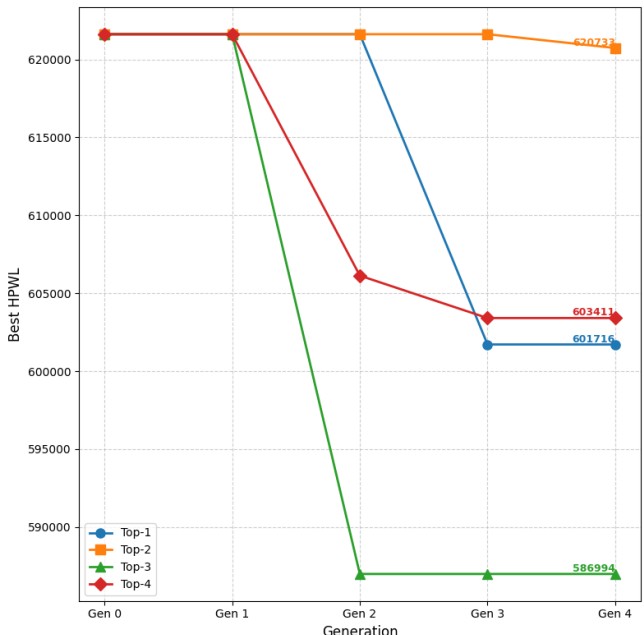

*Figure 3.* Experimental Analysis Results for Parameter Top-K.

*Table 12.* Ablation results of OrderPlace using different LLM backends.

|          | Claude Sonnet 4.5 | DeepSeek-V3.2 | GPT-4o |
|----------|-------------------|---------------|--------|
| adaptec1 | 5.75              | 5.76          | 5.87   |
| adaptec3 | 57.98             | 58.15         | 58.66  |

The results show that different LLMs tend to discover qualitatively different placement-order strategies. For example, Claude Sonnet 4.5 more frequently identifies strategies related to clustering and adaptive gravity, while GPT-4o and DeepSeek-V3.2 discover more strategies associated with field potential, entropy, spring potential, and hybrid potential. This observation indicates that the search process is not trivially dominated by a single fixed strategy pattern, and that different LLMs can explore diverse regions of the strategy space. Meanwhile, more capable LLMs generally exhibit stronger preference understanding and generate effective strategies earlier, leading to faster convergence and higher elite population update rates in the early generations.

Despite the differences in generated strategies and convergence speed, the final best HPWL values achieved by different LLMs remain comparable, as shown in Table 12. This suggests that OrderPlace is not overly sensitive to a specific LLM backend. Instead, multiple distinct strategy-generation trajectories can reach competitive placement quality. Furthermore, the convergence curves show that the best proxy HPWL improves rapidly in the first few generations and then becomes stable after approximately four generations. Extending the search to six generations brings only marginal additional improvement, indicating that the search has largely converged under the current setting. These results further demonstrate that macro placement order is a rich and actionable optimization dimension, and that the proposed LLM-guided search framework is robust across different LLM backends.

## H. Cost, Efficiency, and Scalability

Although OrderPlace introduces LLM-guided evolutionary search, its overall cost remains practical because the LLM is only used to generate candidate placement-order strategies, while strategy evaluation is performed by an efficient proxy mechanism. To quantify the overhead of the complete evolution process, we analyze the LLM API calls, greedy probing time, and end-to-end runtime across different LLM backends and benchmarks. The detailed results are reported in Tables 13, 14, and 15.

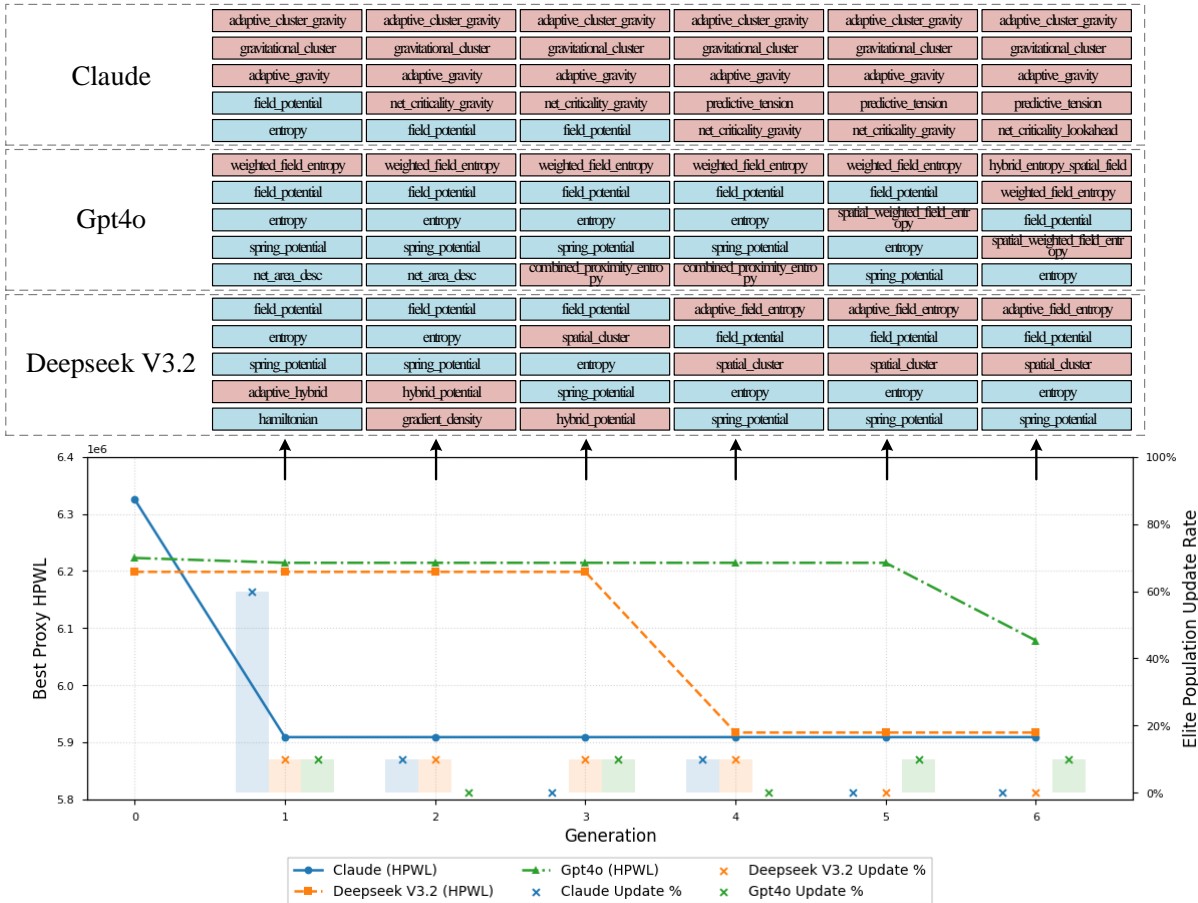

*Figure 4.* Below subfigure is the convergence curve of the best proxy HPWL as the generation number increases, where the bar chart represents the elite population update rate for each generation. Above subfigure is the composition of the elite population for each generation, with red representing the strategies found by the LLM and blue representing the initial strategies.

As shown in Table 13, the number of LLM calls is small and fixed in our setting. For each benchmark and LLM backend, the evolution process requires only 12 API calls, resulting in tens of thousands of tokens rather than hundreds of thousands or millions. For example, on the largest benchmark Adaptec3, GPT-4o consumes 37,109 tokens in total, with an LLM inference time of only 169.7 seconds. This indicates that the LLM-related overhead is minor compared with the overall optimization process. Moreover, since OrderPlace bootstraps the search with manually designed initial strategies instead of generating all strategies from scratch, the framework avoids excessive trial-and-error interactions with the LLM.

The dominant runtime cost comes from evaluating generated strategies through greedy probing. As summarized in Table 14, the per-probe time remains moderate even on Adaptec3. For GPT-4o-generated strategies, the per-probe time ranges from 15.95 seconds to 35.86 seconds; for DeepSeek-V3.2 and Claude Sonnet 4.5, it remains in a similar range. In addition, each generation is bounded by a 1,800-second timeout, which prevents extremely expensive strategies from dominating the search process. Therefore, the evolutionary process has a controllable evaluation budget and does not incur unbounded runtime growth.

Table 15 further reports the end-to-end cost of evolving high-quality strategies. On Adaptec3, the total runtime is approximately 85.6 minutes for GPT-4o, 92.4 minutes for DeepSeek-V3.2, and 111.5 minutes for Claude Sonnet 4.5. The estimated LLM API cost remains very low: approximately $0.19 for GPT-4o, $0.07 for DeepSeek-V3.2, and $0.54 for Claude Sonnet 4.5. These results show that the monetary cost of LLM-guided evolution is negligible compared with the cost of full placement optimization or training-based methods. In particular, unlike reinforcement-learning-based placers such as MaskPlace, which require model training and repeated policy updates, OrderPlace only evolves reusable placement-order strategies through a small number of LLM calls and lightweight proxy evaluations. As a result, its total time cost is comparable to, and in many cases lower than, the training time of standard RL-based placement models.

*Table 13.* LLM API Call Cost Analysis across Different Models and Benchmarks.

| Benchmark | LLM Model | Total Calls | Prompt Tokens | Completion Tokens | Total Tokens | Total Inference Time (s) | Avg. Inference Time (s/call) |
|---|---|---|---|---|---|---|---|
| Adaptec1 | GPT-4o | 12 | 23,784 | 9,052 | 32,836 | 201.8 | 16.8 |
| Adaptec1 | DeepSeek | 12 | 30,340 | 19,363 | 49,703 | 830.1 | 69.2 |
| Adaptec1 | Claude | 12 | 35,882 | 20,766 | 56,648 | 357.0 | 29.8 |
| Adaptec3 | GPT-4o | 12 | 26,530 | 10,579 | 37,109 | 169.7 | 13.9 |
| Adaptec3 | DeepSeek | 12 | 32,866 | 19,936 | 52,802 | 802.2 | 66.9 |
| Adaptec3 | Claude | 12 | 10,003 | 25,840 | 35,842 | 263.8 | 22.0 |

*Table 14.* Strategy Evaluation (Greedy Probing) Time Analysis for Selected High-Quality Strategies.

| Benchmark | LLM Model | Strategy | Greedy Probing Total (s) | Per-Probe Time (s) |
|---|---|---|---|---|
| Adaptec1 | GPT-4o | hybrid_entropy_field | 315.5 | 6.31 |
| | | entropy | 234.2 | 4.68 |
| | | field_potential | 217.2 | 4.36 |
| | | adaptive_hybrid | 502.0 | 10.04 |
| Adaptec1 | DeepSeek | progressive_cluster_entropy | 822.9 | 16.45 |
| | | reinforcement_placement | 1,523.0 | 30.46 |
| | | entropy | 214.2 | 4.28 |
| | | field_potential | 232.2 | 4.64 |
| Adaptec1 | Claude | adaptive_gravity | 166.9 | 3.33 |
| | | adaptive_net_criticality | 138.3 | 2.76 |
| | | quantum_cluster | 166.1 | 3.32 |
| | | entropy | 232.2 | 4.64 |
| Adaptec3 | GPT-4o | hybrid_entropy_spatial_field | 797.9 | 15.95 |
| | | weighted_field_entropy | 1,389.6 | 27.79 |
| | | field_potential | 986.6 | 19.73 |
| | | spatial_weighted_field_entropy | 1,793.4 | 35.86 |
| Adaptec3 | DeepSeek | adaptive_field_entropy | 1,353.6 | 27.07 |
| | | field_potential | 975.6 | 19.51 |
| | | spatial_cluster | 1,466.4 | 29.32 |
| | | entropy | 943.6 | 18.87 |
| Adaptec3 | Claude | adaptive_cluster_gravity | 1,613.1 | 32.26 |
| | | gravitational_cluster | 1,580.6 | 31.61 |
| | | adaptive_gravity | 1,659.5 | 33.19 |
| | | predictive_tension | 1,572.9 | 31.45 |

*Table 15.* End-to-End Total Cost Summary. Est. API Cost is calculated based on publicly available pricing: GPT-4o ($2.50/1M input + $10/1M output), DeepSeek-V3 ($0.27/1M input + $1.10/1M output), Claude 3.5 Sonnet ($3/1M input + $15/1M output).

| Benchmark | LLM Model | LLM Inference (s) | Strategy Evaluation (s) | Total Time (s) | Total Time (min) | Total Tokens | Est. API Cost (USD) |
|---|---|---|---|---|---|---|---|
| Adaptec1 | GPT-4o | 201.8 | 1,268.9 | **1,470.7** | ~**24.5** | 32,836 | ~$0.16 |
| Adaptec1 | DeepSeek | 830.1 | 2,792.3 | **3,622.4** | ~**60.4** | 49,703 | ~$0.07 |
| Adaptec1 | Claude | 357.0 | 703.5 | **1,060.5** | ~**17.7** | 56,648 | ~$0.85 |
| Adaptec3 | GPT-4o | 169.7 | 4,967.5 | **5,137.2** | ~**85.6** | 37,109 | ~$0.19 |
| Adaptec3 | DeepSeek | 802.2 | 4,739.2 | **5,541.4** | ~**92.4** | 52,802 | ~$0.07 |
| Adaptec3 | Claude | 263.8 | 6,426.1 | **6,689.9** | ~**111.5** | 35,842 | ~$0.54 |

# I. End-to-End PPA Evaluation with OpenROAD

Since the ISPD 2005 benchmarks do not provide the required technology and parasitic files for complete PPA evaluation, and commercial back-end tools such as Cadence Innovus and Synopsys ICC2 are not available in our environment, we further evaluate OrderPlace on the ChipBench benchmark using the open-source OpenROAD flow. As shown in Table 16, OrderPlace achieves lower HPWL and congestion on all evaluated designs, and these improvements are also reflected in post-routing metrics. Compared with EfficientPlace, OrderPlace obtains lower routed wirelength on all benchmarks, improves WNS on three benchmarks, improves TNS on three benchmarks, reduces the number of violating paths on three benchmarks, and achieves smaller area on all benchmarks. Although power is slightly higher in some cases, the overall results demonstrate that OrderPlace is not only effective in HPWL optimization, but also competitive in end-to-end PPA quality, including routing, timing, congestion, and area.

*Table 16.* Comparisons of PPA metrics. These metrics include the routed wirelength (WL, um) and power consumption (Power, nW), where smaller values indicate better performance. In contrast, the worst negative slack (WNS, ns) and total negative slack (TNS, ns) are the higher the better, reflecting timing performance, with the best result of each metric for each benchmark in bold.

| Benchmark | Method | Intermediate Metrics | | PPA Metrics | | | | | |
|---|---|---|---|---|---|---|---|---|---|
| | | HPWL ↓ | Congestion ↓ | WL ↓ | Power ↓ | WNS ↑ | TNS ↑ | NVP ↓ | Area ↓ |
| ariane133 | EfficientPlace | 6343114 | 0.304 | 9516763 | **0.352** | -0.964 | -2013.650 | 3360 | 366878 |
| | OrderPlace | **5838502** | **0.299** | **7921340** | 0.368 | **-0.917** | **-1942.530** | **3298** | **366627** |
| ariane136 | EfficientPlace | 9966452 | 0.425 | 13005699 | 0.410 | -1.024 | **-610.876** | **2817** | 397814 |
| | OrderPlace | **9497615** | **0.415** | **12297241** | 0.418 | **-0.693** | -616.743 | 2860 | **397456** |
| dft68 | EfficientPlace | 5307768 | 0.480 | 7050761 | 0.525 | -2.397 | -417.067 | 442 | 96368 |
| | OrderPlace | **5161437** | **0.475** | **6986823** | 0.531 | **-2.208** | **-400.185** | **435** | **96041** |
| wrapper43 | EfficientPlace | 25689796 | 1.331 | 32891409 | 0.464 | **-11.671** | -31150.4 | 9146 | 251324 |
| | OrderPlace | **25162108** | **1.325** | **32738131** | 0.481 | -13.387 | **-30206.8** | **9136** | **250822** |

# J. Ablation Study on the Contribution of LLM

To isolate the source of improvement in OrderPlace, we conduct an ablation study by removing the LLM from the framework and evaluating only a set of manually designed initial placement-order strategies. As shown in Table 17, the LLM-generated strategies consistently outperform all hand-crafted strategies across the evaluated benchmarks. For example, on adaptec3 and adaptec4, the best manually designed strategies achieve HPWL values of 59.00 and 55.80, respectively, while the LLM-discovered strategies reduce them to 52.81 and 48.31. Similar improvements can also be observed on bigblue3, where the LLM-generated strategy significantly reduces HPWL from 66.59 to 35.22. These results indicate that the performance gain does not only come from the proxy-guided evolutionary search or the downstream greedy placement engine. Instead, the LLM plays a key role in discovering more effective and non-trivial placement-order strategies beyond manually designed heuristics. In addition, the large performance variance among different initial strategies shows that the placement-order optimization space is rich and has been insufficiently explored in previous studies.

*Table 17.* Comparison of best HPWL ($\times 10^5$) for ablation experiments on placement order strategies, where inf indicates that there is no valid placement result, and desc and aes represent descending and ascending order, respectively. The best results are marked in **bold**, and the second-best result is underlined.

| | + Net Area Desc | + degree area desc | + area degree desc | + degree desc | + area desc | + degree asc | + area asc | + random | + net area asc | + LLM Design |
|---|---|---|---|---|---|---|---|---|---|---|
| adaptec1 | 5.91 | 5.92 | 6.00 | 6.13 | 6.17 | 9.99 | 10.23 | 10.29 | 11.21 | **5.68** |
| adaptec2 | 51.16 | inf | 39.50 | 43.47 | 39.03 | inf | inf | inf | inf | **28.66** |
| adaptec3 | 59.00 | 62.52 | 63.53 | 66.26 | 64.98 | inf | inf | 112.32 | inf | **52.81** |
| adaptec4 | 60.45 | 57.78 | 58.82 | 56.86 | 55.80 | 84.26 | 82.62 | 87.96 | 83.37 | **48.31** |
| bigblue1 | 2.14 | 2.17 | 2.18 | 2.40 | 2.18 | 2.40 | 2.42 | 5.01 | 2.36 | **1.97** |
| bigblue3 | 67.61 | 72.06 | 111.05 | 66.59 | 121.71 | inf | inf | inf | inf | **35.22** |

