# OpenReview forum: "Order Matters: Unveiling the Hidden Impact of Macro Placement Sequences via Proxy-Guided LLM Evolution"
_ICML.cc/2026/Conference — ICML 2026 regular_

### Official Review · Reviewer_zrAZ · 2026-02-25

**Soundness:** 3
**Presentation:** 2
**Significance:** 3
**Originality:** 2
**Overall Recommendation:** 4
**Confidence:** 1

**Summary:**

This paper focuses on macro placement, a core step in modern chip physical design, addressing key limitations of existing methods: these approaches treat macro placement sequences merely as a static preprocessing step relying on manual heuristics, where suboptimal early sequencing decisions trigger irreversible domino effects that constrain the solution space, coupled with the prohibitively high computational cost of sequence evaluation. It systematically explores the optimization potential of macro placement sequences and proposes an innovative framework to achieve the automatic evolution and efficient optimization of sequencing strategies.
To tackle these issues, the authors present the OrderPlace framework, which leverages proxy-guided large language model (LLM) evolution to automatically discover macro placement sequencing strategies: 1) Theoretically demonstrating the asymptotic degradation of placement quality caused by suboptimal ordering in chain and star topologies, thereby establishing the necessity of sequence optimization; 2) Designing a lightweight proxy evaluation mechanism that efficiently filters candidate sequences via deterministic greedy probing, avoiding the high computational cost of full-scale evaluation; 3) Establishing an LLM-driven strategy evolution system, which takes static and dynamic heuristic strategies as the initial population, generates code-level generalizable sequencing strategies via LLMs, and selects high-quality strategies through evolutionary iteration; 4) Integrating a wire-mask-guided greedy placement process that combines evolved sequencing strategies with locally optimal placement to achieve global layout optimization.
On the standard ISPD 2005 benchmarks, OrderPlace demonstrates significant performance advantages. Additionally, the proxy evaluation mechanism effectively screens high-quality strategies, substantially reducing computational overhead.

**Compliance With Llm Reviewing Policy:**

Affirmed.

**Final Justification:**

My final recommendation is weak accept.The authors’ rebuttal has addressed my concerns, and I will keep my original score.

**Key Questions For Authors:**

1. This paper only adopts claude-sonnet-4.5 as the core LLM backend, without comparing the differences in strategy generation quality, efficiency, and stability among different LLMs , nor explaining the consistency of high-quality strategies generated by LLMs under the same input. Have the authors supplemented multi-LLM comparison experiments? If the performance of strategies generated by different LLMs varies significantly, are there robustness optimization schemes adapted to different models?
2. This paper verified the correlation between proxy evaluation and HPWL, but did not explore it in larger-scale macro placement scenarios. In scenarios with a larger number of macros, does the correlation between proxy evaluation and full-process optimization still remain stable?

**Limitations:**

yes

**Strengths And Weaknesses:**

Soundness: This paper is solidly supported by theory, mathematically proving the asymptotic impact of placement sequences via theorems on chain and star topologies. It adopts standard benchmarks and diverse comparison methods, with comprehensive statistical validation and extended experiments; the proxy evaluation mechanism is verified reliable through experiments, and the research limitations are objectively disclosed. Nevertheless, the verification of LLM model dependence and generalizability in complex scenarios is insufficient, and some mathematical components of the strategies lack adequate theoretical explanations, resulting in limited interpretability.
Presentation: This paper follows an academic paradigm with a clear structure and coherent logic. Methodological details, experimental parameters, and appendix supplements are thorough, ensuring high reproducibility. Related work is accurately positioned with distinct differences from the proposed approach. However, basic explanations of professional terms are lacking, and the background of the interdisciplinary field is insufficiently contextualized, leading to a high understanding threshold for non-EDA readers. Additionally, there are typos and formatting issues in some expressions, increasing the reading burden.
Significance: This paper focuses on key pain points in chip macro placement. The proposed automated sequence optimization scheme has clear industrial value, pioneering the interdisciplinary research direction of "LLM+EDA" and deepening the understanding of placement optimization.
Originality: The work is innovative in its perspective of treating "placement sequence as an independent optimization dimension," creatively combining LLMs, evolutionary algorithms, and proxy evaluation to discover a dynamic multi-factor strategy paradigm adapted to circuit topologies. Nevertheless, some components draw on existing technologies, the demonstration of differences from related work is not thorough enough, and no new tasks or evaluation indicators are proposed. The innovation is more concentrated on perspectives and combinations rather than breakthroughs in brand-new methodologies.

---

> ### Author Rebuttal · Authors · 2026-03-30
>
> **We sincerely thank the reviewer for the detailed and constructive feedback. Below we address each point.**
>
> ## Response to Weakness1: LLM Dependency, Generalization, and Interpretability
>
> We acknowledge the reviewer's concern. Regarding LLM dependency and generalization, please see our detailed multi-LLM comparison in the response to **Q1** below. Regarding the interpretability of mathematical components in discovered strategies, we note that this is an inherent characteristic of LLM-generated code; however, compared to RL/DL methods that encode policies in neural networks with millions of opaque parameters, OrderPlace produces **readable Python code**, which already offers significantly higher interpretability. We will add further annotation and theoretical discussion of key mathematical terms in the revised manuscript.
>
> ## Response to Weakness2: Accessibility for Non-EDA Readers
>
> We thank the reviewer for pointing out this accessibility issue. This paper aims to explore the intersection of machine learning and electronic design automation (EDA), and we fully agree that it is essential for the broader ML community to understand its content. We will include basic explanations of EDA-specific technical terms and enhance the background introduction for cross-disciplinary readers in the revised version. All typographical errors and formatting issues will also be corrected.
>
> ## Response to Weakness3: Novelty Relative to Existing Techniques
>
> We respectfully clarify that OrderPlace introduces several **methodological innovations** beyond combining existing components:
>
> 1. **Dynamics-inspired initial strategies.** To enhance the LLM's test-time preference for the macro placement problem, we design a suite of dynamics-inspired dynamic ordering strategies as evolutionary seeds.
> 2. **Comparative prompting for strategy generation.** Unlike EoH-style methods that use the LLM as a mutation operator, OrderPlace guides the LLM to **directly generate strategies** through comparative questioning inspired by the DPO literature, effectively reducing ineffective mutations.
> 3. **Proxy evaluator.** To tackle the expensive evaluation bottleneck in ordering strategy search, we propose a lightweight proxy evaluator that significantly accelerates the evolutionary loop.
>
> More importantly, **OrderPlace's primary contribution lies in shifting the optimization perspective** of macro placement toward *placement order*—a factor largely overlooked by the community. In many prior works, the placement order strategy is not even documented; we had to examine their source code to identify how ordering was handled. We believe this insight itself constitutes a significant contribution.
>
> ## Response to Question1: Multi-LLM Comparison
>
> We conducted a case study on Adaptec3 using three LLMs—**Claude Sonnet 4.5, GPT-4o, and DeepSeek-V3**—with evolutionary generations extended to 6. The convergence curves and discovered strategies are shown below.
>
> - https://anonymous.4open.science/r/OrderPlace/convergence_curves.png
> - https://anonymous.4open.science/r/OrderPlace/Different_LLMs_in_OrderPlace.png
>
> **Key findings:**
> - Different LLMs discover **qualitatively different strategies**, demonstrating that the framework is not trivially tied to a single model.
> - More capable models exhibit stronger preference understanding, discovering elite strategies earlier and converging faster.
> - Despite these differences, the **best HPWL achieved remains comparable** across LLMs, indicating that the optimization space of macro placement order is substantial and directly impactful—the framework is robust to LLM choice.
>
> ## Response to Question2: Proxy Evaluator Robustness at Scale
>
> We have validated the proxy evaluator on larger-scale benchmarks (adaptec3 and bigblue3). As shown below, the proxy metric remains **strongly correlated** with final HPWL, correctly identifying the top strategy in both cases with only minor mid-ranking variations. This confirms the robustness of our proxy-based approach at scale and supports selecting the top-4 strategies based on the proxy metric.
>
> - https://anonymous.4open.science/r/OrderPlace/Validation_Evaluator_in_adaptec3.png
> - https://anonymous.4open.science/r/OrderPlace/Validation_Evaluator_in_bigblue3.png
>
> ---
>
> *We will incorporate all clarifications and additional results into the revised manuscript. We thank the reviewer again for the valuable feedback.*

---

> > ### Author Rebuttal · Reviewer_zrAZ · 2026-04-01
> >
> > Thanks to the authors for their explanations.

---

> > > ### Author Response · Authors · 2026-04-01
> > >
> > > Dear Reviewer,
> > >
> > > Thank you very much for your acknowledgment that the concerns have been fully addressed. We truly appreciate your time and feedback.
> > >
> > > We were wondering whether this updated assessment could be reflected in the overall score, if you feel it is appropriate. We completely understand that the final decision is at your discretion.
> > >
> > > Thank you again for your consideration.

---

### Official Review · Reviewer_CoZs · 2026-03-12

**Soundness:** 4
**Presentation:** 3
**Significance:** 3
**Originality:** 3
**Overall Recommendation:** 5
**Confidence:** 3

**Summary:**

This paper demonstrates that the placement sequence in greedy macro placement is a critical but overlooked optimization dimension. The authors provide theoretical bounds showing worst-to-best HPWL ratios of Ω(l) for chain topologies and Θ(g) for star topologies. They propose OrderPlace, which uses LLM-guided evolutionary search to automatically discover code-level ordering strategies, combined with a proxy evaluation mechanism for efficient candidate filtering. Experiments on ISPD2005 benchmarks show OrderPlace achieves best HPWL on 7/8 circuits, outperforming EGPlace by 14.08% and WireMask-EA by 34.04%.

**Compliance With Llm Reviewing Policy:**

Affirmed.

**Key Questions For Authors:**

1) How sensitive are the discovered strategies to the choice of LLM? The paper uses Claude Sonnet 4.5—would GPT-4 or DeepSeek-V3 discover qualitatively different strategies?

2) The search space is small (4 iterations × 4 candidates = 16 LLM-generated strategies). Have you analyzed whether the search has converged, or would additional iterations yield further improvements?

3) Can you clarify the methodological distinction from Liu et al. (ICML 2024) beyond the application domain?

**Limitations:**

yes

**Strengths And Weaknesses:**

Strengths

1) The paper introduces a genuine perspective shift: treating placement ordering as an optimizable dimension rather than a fixed heuristic. Theorems 1–2 provide formal backing that ordering can cause asymptotic quality degradation. This insight has lasting value independent of the specific method.

2) 12 baselines spanning analytical, RL, hybrid, and evolutionary methods. Wilcoxon rank-sum tests at 0.05 significance, 5 independent runs, convergence curves, congestion analysis, mixed-size placement, and proxy evaluator validation. This is exemplary experimental rigor.

3) Average rank 1.17 across all circuits, with consistent improvements across diverse designs. Figure 2 shows OrderPlace converges faster than EGPlace on every benchmark. The discovered strategies (gravitational models, entropy reduction, net closure) provide interpretable, transferable design knowledge.

4) The framework is well-documented: complete prompt templates (Appendix C), all 24 discovered strategy formulations (Appendix F), and proxy evaluator validation (Appendix G). This level of transparency strongly supports reproducibility.

Weaknesses

1) The LLM evolutionary search framework closely follows Liu et al. (ICML 2024, "Evolution of Heuristics"). While the application domain and discovered insights are new, the core methodology (LLM generates code → evaluate → evolve) is not original.

2) Only validated on greedy placement. Whether ordering matters equally for stochastic, RL-based, or analytical methods is unknown. The practical scope is thus narrower than the paper’s general claims suggest.

---

> ### Author Rebuttal · Authors · 2026-03-30
>
> **We sincerely thank the reviewer for the insightful comments. Below we address each point.**
>
> ## Response to Weakness1 & Question3: Novelty over EoH (Liu et al., ICML 2024)
>
> We respectfully clarify that OrderPlace introduces several **methodological innovations** beyond a change of application domain:
>
> 1. **Dynamics-inspired initial strategies.** To enhance the LLM's test-time preference for the macro placement problem, we design a suite of dynamics-inspired dynamic ordering strategies as evolutionary seeds, rather than starting from generic heuristics.
> 2. **Strategy generation via comparative prompting.** Unlike EoH-style methods that use the LLM as a mutation operator, OrderPlace guides the LLM to **directly generate new strategies** through preference-based comparative questioning and ranking. This effectively reduces ineffective mutations and improves search efficiency.
> 3. **Proxy evaluator.** To address the expensive evaluation bottleneck in ordering strategy search, we propose a lightweight proxy evaluator that significantly accelerates the evolutionary loop.
>
> More importantly, we wish to emphasize that **OrderPlace's primary contribution lies in shifting the optimization perspective** of macro placement toward *placement order*—a factor that has been largely overlooked or even unmentioned by the community. In many prior works, the placement order strategy is not documented in the paper at all; we had to carefully examine their source code to identify how ordering was handled. We believe this insight itself constitutes a significant and complementary contribution to the field.
>
> ## Response to Weakness2: Validation Limited to Greedy Placement
>
> We acknowledge this limitation and have explicitly discussed it in the **Limitations** section of our paper. Extending the order optimization to all sequential decision-making methods (e.g., RL-based approaches) would indeed broaden the contribution, and we have explored but not yet achieved a unified framework.
>
> That said, we argue that using greedy placement is a **deliberate and well-motivated design choice**. As discussed in our paper, macro placement outcomes are jointly determined by placement *order* and placement *strategy*. If the placement strategy itself is learnable (e.g., via RL), it becomes extremely difficult to disentangle the individual effect of placement order on the final result. By fixing the placement strategy to a greedy policy, we **maximize the ability to isolate and analyze the impact of placement order** on layout quality.
>
> We firmly believe that OrderPlace makes a positive contribution to the community by drawing attention to this overlooked optimization dimension.
>
> ## Response to Question1 & Question2: LLM Sensitivity and Search Convergence
>
> We conducted a case study on the Adaptec3 benchmark using three LLMs—**Claude Sonnet 4.5, GPT-4o, and DeepSeek-V3**—with the number of evolutionary generations extended to 6. The convergence curves, elite population update rates, and discovered strategies are shown in the figures below.
>
> - https://anonymous.4open.science/r/OrderPlace/convergence_curves.png
> - https://anonymous.4open.science/r/OrderPlace/Different_LLMs_in_OrderPlace.png
>
> **Key findings:**
> - Different LLMs discover **qualitatively different strategies**, demonstrating that the search is not trivially dominated by a single solution pattern.
> - More capable models exhibit stronger preference understanding, discovering elite strategies earlier and converging faster.
> - Despite these differences, the **best HPWL achieved across LLMs remains comparable**, suggesting that the optimization space of macro placement order is substantial and directly impactful—multiple distinct strategies can reach near-optimal quality.
>
> This further validates that placement order is a rich and actionable optimization dimension, rather than a marginal factor.
>
> ---
>
> *We will incorporate all clarifications into the revised manuscript. We thank the reviewer again for the valuable feedback.*

---

> > ### Author Rebuttal · Reviewer_CoZs · 2026-04-04
> >
> > Thanks for your great answers! I will remain my score.

---

> > > ### Author Response · Authors · 2026-04-06
> > >
> > > Thanks~
> > >
> > > :)

---

### Official Review · Reviewer_6eCj · 2026-03-12

**Soundness:** 3
**Presentation:** 3
**Significance:** 2
**Originality:** 3
**Overall Recommendation:** 4
**Confidence:** 4

**Summary:**

This paper studies macro placement ordering, and its core claim is that placement order is a major and under-optimized factor in final placement quality. To exploit this, the paper proposes OrderPlace, which uses an LLM-driven evolutionary search over code-level ordering strategies together with a lightweight proxy evaluator that filters candidates before running the full greedy placement procedure. On ISPD 2005 benchmarks, the paper reports stronger HPWL than a broad set of prior methods, including recent RL and hybrid baselines.

**Compliance With Llm Reviewing Policy:**

Affirmed.

**Ethical Review Concerns:**

This paper focuses on macro placement ordering as an underexplored optimization dimension and uses LLM-guided evolutionary search to discover effective strategies. My initial concerns focused on isolating the LLM’s contribution, proxy evaluator validity, and cross-design generalization. The rebuttal addressed most of my concerns. The ablation experiments clearly isolate the LLM’s contribution over hand-crafted strategies, and the proxy evaluator is well-validated on larger benchmarks.
While I still doubt this approach will not be realistic in the real design scenarios, the core insight is original and interesting. I have adjusted my score accordingly.​​​​​​​​​​​​​​​​

**Final Justification:**

This paper focuses on macro placement ordering as an underexplored optimization dimension and uses LLM-guided evolutionary search to discover effective strategies. My initial concerns focused on isolating the LLM’s contribution, proxy evaluator validity, and cross-design generalization. The rebuttal addressed most of my concerns. The ablation experiments clearly isolate the LLM’s contribution over hand-crafted strategies, and the proxy evaluator is well-validated on larger benchmarks.
While I still doubt this approach will not be realistic in the real design scenarios, the core insight is original and interesting. I have adjusted my score accordingly.​​​​​​​​​​​​​​​​

**Key Questions For Authors:**

1. How strong is the correlation between the proxy evaluator and final full-placement quality across the benchmark suite?

2. How much of the gain comes from the LLM-generated strategies themselves, versus the general proxy-guided evolutionary search procedure? What is the exact benefit of having LLM in the loop? Is it able to summarize any general recipes for picking the sequence?

**Limitations:**

yes

**Strengths And Weaknesses:**

The paper presents an unexplored interesting idea, treating macro placement order as an optimization target rather than a fixed heuristic. The full pipeline also combines a proxy evaluator to enable fast exploration. The evaluation results are strong. The paper reports the best HPWL on 7 of 8 ISPD 2005 circuits. However, my concern is that the paper does not yet isolate its main source of improvement clearly enough. It remains unclear how much of the gain should be attributed to the LLM itself, as opposed to the surrounding proxy-guided evolutionary search procedure and the downstream greedy placement engine. This work is more convincingly a search framework than a clean validation of LLM-based strategy discovery. Also, it is difficult to judge whether the discovered strategies generalize to newer designs or to settings where the proxy evaluator is a poorer approximation of final placement quality. Finally, while the paper compares against many baselines, the comparison budgets and implementation details need to be adjusted carefully. For example, the paper uses parallel strategy evaluation and a fixed LLM-generation loop, while several baseline numbers are taken from prior papers. Overall, the paper is a good systems-level search framework with a fresh look at the placement problem, the exact benefit of the LLM requires further validation.

---

> ### Author Rebuttal · Authors · 2026-03-30
>
> **We sincerely thank the reviewer for the rigorous and constructive feedback. Below we address each point.**
>
> ## W1 & Q2: Isolating the Source of Improvement
>
> We first clarify that the greedy proxy does **not** participate in the final macro placement process—it serves solely as a lightweight evaluator to assess the quality of ordering strategies.
>
> To explicitly isolate the contribution of the LLM, we conducted ablation experiments by removing the LLM and evaluating a set of manually designed initial strategies alone. As shown in the result (https://anonymous.4open.science/r/OrderPlace/Ablation_Experiments_on_Placement_Order_Strategies.png), the performance gap between LLM-discovered strategies and the best hand-crafted ones is **substantial**. Furthermore, the large variance among different initial strategies confirms that the optimization space of placement order is vast—a point that has been largely overlooked by the community.
>
> ## W2: Generalization of Discovered Strategies
>
> As shown in the supplementary tables below, OrderPlace's time and token costs are modest. Combined with the analysis in Appendix F10 and Table 8, each circuit design exhibits **distinct strategy preferences**, suggesting that OrderPlace is best suited for discovering design-specific strategies rather than a single universal rule.
>
> We emphasize that the primary contribution of this work is **not** generalization across designs, but rather the **first systematic revelation** that macro placement order is an important yet neglected optimization dimension deserving dedicated attention.
>
> - https://anonymous.4open.science/r/OrderPlace/LLM_API_Call_Analysis.png
> - https://anonymous.4open.science/r/OrderPlace/End_to_end_total_cost_analysis.png
>
> ## W3: Comparison Budget and Implementation Details
>
> We will supplement the final version with more detailed analyses of computational cost, token usage, and implementation specifics. Please also refer to our responses to Reviewer 1 (Cfrf) W3, Q1, and Q3 for comprehensive cost breakdowns.
>
> ## Q1: Proxy Evaluator Correlation with Final Placement Quality
>
> To validate the proxy evaluator, we provide a verification experiment in Appendix G.1. Although our framework typically performs full optimization only on the evaluator's top-4 ranked strategies, we additionally ran full optimization on **all** generated strategies to verify whether the evaluator's ranking aligns with actual performance. The results confirm that the evaluator accurately distinguishes the "elite tier" from other strategies.
>
> We have also validated the proxy evaluator on larger-scale benchmarks. As shown below, the proxy metric remains **strongly correlated** with final HPWL, correctly identifying the top strategy in both cases.
>
> - https://anonymous.4open.science/r/OrderPlace/Validation_Evaluator_in_adaptec3.png
> - https://anonymous.4open.science/r/OrderPlace/Validation_Evaluator_in_bigblue3.png
>
> ---
>
> *We hope the above responses adequately address the reviewer's concerns and welcome any further questions. In addition to the points discussed above, we have conducted several supplementary experiments, summarized below:*
>
> - **PPA evaluation on ChipBench using OpenROAD**, validating OrderPlace's competitiveness beyond HPWL (https://anonymous.4open.science/r/OrderPlace/Performance_on_PPA_Metrics.png)
> - **LLM API call analysis**, detailing token consumption per evolutionary generation (https://anonymous.4open.science/r/OrderPlace/LLM_API_Call_Analysis.png)
> - **End-to-end cost analysis**, showing total time and monetary cost across benchmarks (https://anonymous.4open.science/r/OrderPlace/End_to_end_total_cost_analysis.png)
> - **Greedy probing time analysis**, confirming scalability with bounded per-probe runtime (https://anonymous.4open.science/r/OrderPlace/Greedy_probing_time_analysis.png)
> - **ICCAD 2015 evaluation**, demonstrating consistent state-of-the-art results on additional benchmarks (https://anonymous.4open.science/r/OrderPlace/ICCAD(Superblue)_resutl.png)
> - **Ablation on placement order strategies**, isolating the LLM's contribution from initial strategies (https://anonymous.4open.science/r/OrderPlace/Ablation_Experiments_on_Placement_Order_Strategies.png)
> - **Multi-LLM convergence comparison**, analyzing search behavior across Claude Sonnet 4.5, GPT-4o, and DeepSeek-V3 (https://anonymous.4open.science/r/OrderPlace/convergence_curves.png)
> - **Strategy diversity across LLMs**, showing qualitatively different yet comparably effective strategies (https://anonymous.4open.science/r/OrderPlace/Different_LLMs_in_OrderPlace.png)
> - **Proxy evaluator validation at scale**, confirming strong correlation on Adaptec3 and Bigblue3 (https://anonymous.4open.science/r/OrderPlace/Validation_Evaluator_in_adaptec3.png, https://anonymous.4open.science/r/OrderPlace/Validation_Evaluator_in_bigblue3.png)
>
> *We will incorporate all additional results and clarifications into the revised manuscript. We thank the reviewer again for the valuable feedback.*

---

> > ### Author Rebuttal · Reviewer_6eCj · 2026-04-02
> >
> > Thanks for the further clarification. I will adjust my score.

---

> > > ### Author Response · Authors · 2026-04-08
> > >
> > > Thank you for your understanding and for adjusting the score. We sincerely appreciate your time and consideration.
> > >
> > > :)

---

### Official Review · Reviewer_Cfrf · 2026-03-13

**Soundness:** 2
**Presentation:** 2
**Significance:** 2
**Originality:** 3
**Overall Recommendation:** 4
**Confidence:** 4

**Summary:**

This paper addresses the ordering issue in macro placement by investigating how the sequence of placement affects the final design quality. The authors propose OrderPlace, a framework that leverages Large Language Models (LLMs) to automatically evolve and generate macro ranking strategies in the form of Python code. To overcome the high computational cost of evaluating placement quality during the evolution process, the paper introduces a proxy-guided evaluation mechanism using a lightweight greedy probe.

**Compliance With Llm Reviewing Policy:**

Affirmed.

**Final Justification:**

I am upgrading my final recommendation to Weak Accept. The authors’ rebuttal has resolved my initial major concerns, particularly by providing more experimental results on broader benchmarks with comparisons to more SOTA methods, which validate their method’s performance. Given the strengthened technical justification and empirical evidence, I am raising my evaluation.

**Key Questions For Authors:**

1. The authors may consider running experiments on more recent benchmarks such as ICCAD 2015, Superblue, or 7nm/5nm designs. Specially, in the industrial design of more macro units, what is the single-run time of the greedy probe? Will the evolutionary process face a dimensionality explosion as the design scale increases?
2. If the evolved ordering is used to initialize an analytical placer like DREAMPlace instead of a sequential placer, is there still a significant improvement in final wirelength? Or does the analytical solver's global optimization negate the advantages of the initial sequence?
3. What is the total estimated cost (in terms of time and LLM API fees) to evolve a high-quality strategy for a single design? How does this compare to the training time of a standard RL model like MaskPlace?
4. Have the authors verified any of the ISPD 2005 results through a full-flow routing and timing analysis tool? HPWL improvements do not always correlate with better timing closure.

**Limitations:**

No. The current limitation of this study is that it is only effective for sequence-sensitive greedy placers and does not explore collaborative interactions with random search or large-scale deep learning models. In addition, since it has only been validated on ISPD 2005, its performance under modern heterogeneous integration and advanced packaging constraints remains unknown.

**Strengths And Weaknesses:**

**Strengths**

1. The paper shifts the focus from spatial coordinate optimization (where to place) to sequence optimization (in what order to place), and provides both theoretical proofs (Chain/Star topologies) and empirical evidence.
2. Using LLMs to evolve *executable code* for ranking strategies is a sophisticated approach.

**Weaknesses**

1. The evaluation is strictly limited to the ISPD 2005 dataset. These benchmarks are nearly two decades old and consist of small-scale designs that do not reflect the complexity, macro density, or constraints of modern industrial chips. This significantly undermines the claim of "state-of-the-art" performance in a modern context.
2. The results rely almost exclusively on HPWL. There is no closed-loop validation using industrial back-end tools (e.g., Cadence Innovus or Synopsys ICC2) to report critical metrics like TNS/WNS (Timing), Routing Congestion, or Total Power, which are the true measures of success in chip design.
3. Although a proxy mechanism is introduced in this paper, the article lacks a detailed analysis of the total time overhead and token cost of the entire LLM evolution process (including multiple API calls and hundreds of iterative generations), so the cost in practical industrial production environments remains unclear.
4. From a practical perspective, algorithms generated by LLMs often contain complex nonlinear terms and multiple parameter couplings. Although this black-box code performs well, it is very difficult for backend chip engineers to manually fine-tune or troubleshoot it.

---

> ### Author Rebuttal · Authors · 2026-03-30
>
> **We sincerely thank the reviewer for the thorough and constructive comments. Below, we address each weakness and question in detail.**
>
> ## Response to Weakness 1: Benchmark Limited to ISPD 2005
>
> We appreciate the reviewer's concern regarding the age of the ISPD 2005 benchmarks. We would like to respectfully clarify our rationale for choosing this dataset:
>
> 1. **Explicit macro specification.** Unlike many publicly available benchmarks, the ISPD 2005 dataset explicitly designates macros within its design files. This is a critical requirement for rigorously studying the macro placement problem, as it eliminates ambiguity in problem formulation.
> 2. **Largest available macro scale.** The ISPD 2005 benchmarks contain 543–1,329 macros per design, which remain among the largest publicly available macro-level benchmarks to date.
> 3. **Why not ICCAD 2015 (Superblue)?** The ICCAD 2015 dataset does not explicitly specify macros. Existing works typically resort to heuristic selection—sorting cells by area in descending order and designating the top-N (e.g., the top 512, or cells exceeding 10× the average area) as macros. We are concerned that such ad hoc practices may compromise the diversity and authenticity of the macro placement problem, thereby undermining a faithful analysis of the impact of macro placement order.
> 4. **Community consensus.**  ISPD 2005 remains the primary benchmark adopted by the latest state-of-the-art works, including EGPlace (ICML 2025), RSPlace (AAAI 2026), BBOPlace (TEVC 2025), and EA-Rotation (TCAD 2025).
>
> Nevertheless, to further strengthen the evaluation, we have additionally conducted experiments on the ICCAD 2015 benchmarks (https://anonymous.4open.science/r/OrderPlace/ICCAD(Superblue)_resutl.png). As shown in the supplementary table below, OrderPlace consistently achieves the best results across all Superblue designs, further validating the generalizability of our approach.
>
> ## Response to Weakness 2 & Question 4: HPWL-Only Evaluation and Lack of PPA Verification
>
> Both ISPD 2005 and ICCAD 2015 lack parasitic parameter files (`.tf`, `.ndm`, `tech lef`, `RC`) required for PPA evaluation, and we do not have access to commercial tools (Cadence Innovus, Synopsys ICC2). To address this, we adopted the ChipBench (ICLR 2025) dataset and performed end-to-end evaluation using the open-source OpenROAD toolchain. Results show that OrderPlace not only achieves significant advantages in wirelength and congestion, but also exhibits strong competitiveness across all PPA metrics (see https://anonymous.4open.science/r/OrderPlace/Performance_on_PPA_Metrics.png).
>
> ## Response to Weakness 3, Question 1 & Question 3: Cost, Efficiency, and Scalability
>
> 1. **Cost & Efficiency:** Our evolutionary process is highly efficient. For the largest benchmark Adaptec3, the end-to-end cost is only ~$0.19 (GPT-4o) with a total runtime of ~80 minutes, comparable to or faster than training RL models like MaskPlace. We bootstrap evolution with well-designed initial strategies rather than generating from scratch, maintaining low overhead (see https://anonymous.4open.science/r/OrderPlace/LLM_API_Call_Analysis.png and https://anonymous.4open.science/r/OrderPlace/End_to_end_total_cost_analysis.png).
> 2. **Scalability:** Greedy probing time for Adaptec3 remains at 15.9–35.8 seconds per probe. Each generation is bounded by a 1,800-second timeout, with 4 strategies evaluated in parallel over 4 generations(`No need for hundreds of iterations`), yielding a maximum runtime ≤ 120 minutes (https://anonymous.4open.science/r/OrderPlace/Greedy_probing_time_analysis.png).
>
> ## Response to Weakness 4: Interpretability of LLM-Generated Strategies
>
> Each algorithm has its own unique strengths, and we believe there should be no bias in this regard. At the same time, we argue that the inability to support interactive decision-making is not a limitation unique to OrderPlace. For reinforcement learning and deep learning methods, their policies are encoded in deep neural networks with millions of parameters, which are essentially a complete black box to engineers.
>
> In contrast, OrderPlace generates human-readable Python code (code-level strategies). Compared with neural network weights or pure coordinate outputs, this representation offers significantly higher interpretability and operability.
>
> ## Response to Question 2: Using Evolved Order to Initialize Analytical Placers (e.g., DREAMPlace)
>
> DREAMPlace-style analytical placers do not involve sequential placement. Their core mechanism is simultaneous global optimization—all cell coordinates are updated concurrently via gradient descent. Therefore, the concept of "placement order" is not applicable.
>
> ---
>
> *We hope the above responses adequately address the reviewer's concerns. We will incorporate all additional experimental results and clarifications into the revised manuscript. We sincerely thank the reviewer again for the valuable feedback that has helped strengthen our work.*

---

> > ### Author Rebuttal · Reviewer_Cfrf · 2026-04-06
> >
> > Thanks for the clarification. I will adjust my score.

---

> > > ### Author Response · Authors · 2026-04-06
> > >
> > > Thank you for your understanding and for adjusting the score. We sincerely appreciate your time and consideration.

---

### Decision · Program_Chairs · 2026-04-30

**Decision:**

Accept (regular)

**Comment:**

This paper studies an important and previously underexplored aspect of macro placement, namely the impact of placement order on final layout quality. Reviewers generally agreed that this perspective is novel and meaningful, and that the proposed OrderPlace framework is technically interesting. In particular, the combination of LLM-guided strategy evolution with a lightweight proxy evaluator was viewed as a practical and effective way to search over ordering strategies, and the empirical results were considered strong.

The main concerns were about benchmark coverage, reliance on HPWL, the cost and scalability of the LLM-driven search process, and the extent to which the gains should be attributed specifically to the LLM rather than the broader search framework. The rebuttal addressed these concerns well by adding stronger validation on newer benchmarks, PPA-oriented evaluation, cost and runtime analysis, multi-LLM comparisons, and clearer ablations isolating the contribution of the learned strategies.

Overall, I find this to be a strong paper with a clear contribution, solid empirical support, and meaningful practical relevance. I therefore recommend accept.